# Increased labor losses and decreased adaptation potential in a warmer world

Luke A. Parsons[1]✉, Drew Shindell [1], Michelle Tigchelaar [2], Yuqiang Zhang [1,3] & June T. Spector [4]

Working in hot and potentially humid conditions creates health and well-being risks that will increase as the planet warms. It has been proposed that workers could adapt to increasing temperatures by moving labor from midday to cooler hours. Here, we use reanalysis data to show that in the current climate approximately 30% of global heavy labor losses in the workday could be recovered by moving labor from the hottest hours of the day. However, we show that this particular workshift adaptation potential is lost at a rate of about 2% per degree of global warming as early morning heat exposure rises to unsafe levels for continuous work, with worker productivity losses accelerating under higher warming levels. These findings emphasize the importance of finding alternative adaptation mechanisms to keep workers safe, as well as the importance of limiting global warming.

[1] Nicholas School of the Environment, Duke University, Durham, NC, USA. [2] Center for Ocean Solutions, Stanford University, Stanford, CA, USA. [3] Department of Environmental Sciences and Engineering, University of North Carolina at Chapel Hill, Chapel Hill, NC, USA. [4] Department of Environmental and Occupational Health Sciences, University of Washington, Seattle, WA, USA. ✉email: luke.parsons@duke.edu

Hundreds of millions of people are already exposed to unsafe levels of heat and potential high humidity every year[1]. Humid heat is particularly dangerous because high ambient temperatures combined with high humidity impede the body's ability to lose body heat to the outside environment by evaporative cooling from sweating[2,3]. In high humidity and temperatures, outdoor workers must slow work, hydrate, and take breaks in the shade to allow the body to cool off and maintain a safe internal body temperature or risk injury, illness, or death if they continue to work at high exertion levels[2,4–7]. Workers in many low-latitude locations already experience heat exposure that makes physical labor unsafe[1,8–10]. Labor productivity losses associated with reductions in work rate due to heat exposure can be as high as ~280–311 billion $US per year[11,12], most of which are due to losses in low- and middle-income countries in heavy manual labor, such as agriculture and construction[11,13,14]. In the coming century, human-driven warming of the planet will push many low-latitude regions even further into uncomfortable and unsafe conditions for outdoor labor[1,14,15], with heat exposure increasing relatively linearly as a function of global temperatures[1,14,16,17].

Basic adaptation measures may help reduce impacts of increasing future heat for outdoor workers. These measures include ensuring adequate hydration, rest breaks in the shade, acclimatization to heat, personal cooling strategies[18], and moving work hours to cooler parts of the day. In locations where regulations designed to protect workers from heat[19] are not in place, workers are already shifting schedules to limit heat exposure[20–24]. Changes in work hours must consider implications for worker health and well-being, competing hazards from shifting work times, and industry-specific aspects. A comprehensive understanding of potential heat exposure and health costs and benefits of shifting work times is needed to weigh trade-offs and to inform decision-making and policies that support adaptation to heat. Previous work has focused on quantifying lost hours due to heat exposure in the 12-h workday[11,25] or on how many work hours would need to be moved to the morning to maintain productivity[26], but so far the feasibility of moving work hours as an adaptation mechanism has not been quantified on a global scale. Here we combine heat exposure estimates from reanalysis data[1] with patterns of warming[16] from the latest climate model projections to examine workshift 'adaptation potential', or what percent of work time is recoverable if laborers move work hours from the hottest hours of the day to cooler hours, and how work loss and adaptation potential change as the planet warms. Although here we define 'adaptation potential' as the ability of workers to shift labor to cooler hours, as mentioned above, adaptation mechanisms are not limited to time shifting.

## Results

### Recent local and global labor losses from heat exposure.
We use simplified Wet Bulb Globe Temperatures (sWBGT), a heat metric calculated from available reanalysis[1] and climate model output[14], to examine heat exposure impacts on outdoor heavy labor productivity (including agriculture, forestry, fisheries, and construction industries, hereafter 'heavy labor'; "Methods"). The advantage of sWBGT, and similar metrics that account for heat and humidity, is that it enables us to interpret both temperature and humidity in relation to heat exposure and heat stress for working individuals. The term 'heat exposure' here refers to conditions that are either simply too hot or are hot and humid enough to cause labor losses; these conditions can include high temperatures with moderate to low humidity, or moderate-to-high temperatures with high humidity, both of which would impact individuals conducting heavy labor. Here we use an exposure response function[13,25] (ERF) that relates heat exposure to labor productivity losses ("Methods"). This ERF shows small (<1%) productivity losses at sWBGT of ~20 °C, 10% losses at ~27 °C, 50% losses at ~32.5 °C, and 90% losses at ~38 °C (Supplementary Fig. 1).

In many low-latitude locations, such as those shown in Fig. 1, heat exposure in the shade is already at or approaching levels that lead to substantial heavy labor productivity losses both in the morning and at midday. For example, in the present-day climate, an average summer day in a location like New Delhi, India or Doha, Qatar exposes workers in the shade to midday heat that would cause productivity losses of ~15–20 min/h of work time. By contrast, the early morning hours tend to still be cool enough to approach 'safe' work thresholds for continuous heavy manual labor, with <10 min/h productivity losses.

By overlaying International Labour Organization (ILO) heavy labor statistics, working-age population data, and estimates of sWBGT and associated hours lost calculated from the European Centre for Medium-Range Weather Forecast 5th Generation Reanalysis (ERA5) data ("Methods"), we estimate global sums of labor lost. In the last two decades (2001–2020), an average of 228 billion hours (±27 billion hours/year) of heavy labor have been lost per year due to heat exposure in the 12-h workday[25], with losses peaking in 2016 (274 billion hours/year) and 2019 (270 billion h/year). These losses are heavily focused in the agriculture sector, which lost ~220 billion hours in 2016 and ~217 billion hours in 2019[13]. Even though many locations are currently cool enough for minimal labor losses early in the day (Fig. 1), there are already about 6 billion lost hours per year globally in the coolest hour of the day alone (Supplementary Fig. 2). Although there is a general upward trend in global labor lost at all hours of the day for the last several decades, global labor losses notably spike during anomalously warm years in the tropical Pacific (i.e., 'El Niño' years: 1982–1983, 1997–1998, 2004–2005, 2009–2010, 2014–2016).

**Warming patterns in climate models.** Coupled Modeling Intercomparison Project, Phase 6 (CMIP6) projections[27] show that future sWBGT will increase relatively linearly as the globe warms[17] ("Methods"). For illustrative purposes, we show the warming patterns for the 75th percentile of daily temperatures, specific humidity, and sWBGT in Fig. 2 (warming patterns from individual CMIP6 models shown in Supplementary Fig. 3). Daily mean, maximum, and minimum temperatures over tropical land areas warm at a rate of ~1–1.2 °C per degree of global warming (Fig. 2a; Supplementary Fig. 4; TextS1). Additionally, CMIP6 models generally agree on the magnitude of specific humidity increases as global temperatures increase, with the fastest increases in humidity across the tropics and the Middle East, South Asia, southern and southeastern Asia, and southeastern North America (Fig. 2b). CMIP6 models also show good agreement in the spatial patterns and magnitudes of local sWBGT changes per degree of global warming (Fig. 2c; Supplementary Figs. 5 and 6). Specifically, between ~40° N and 40° S, zonal means over land show local changes in sWBGT of ~1–1.2 °C per degree of global warming[16], with maxima approaching ~1.5 °C per degree of global warming at low latitudes (Supplementary Fig. 6). If we compare the local magnitude of 2-m air temperature and sWBGT warming patterns, we find that air temperature generally rises faster in regions that experience low specific humidity changes per degree of global warming, whereas sWBGT warms at least as quickly as local temperatures in regions with increases in humidity (Supplementary Fig. 7).

**Future warming and nonlinear labor losses.** We combine the diurnal sWBGT cycle from ERA5 (Fig. 1) with monthly sWBGT warming patterns from CMIP6 models (monthly warming

**Fig. 1 Diurnal cycles of heat exposure and labor productivity losses.** Climatological average (2001-2020) diurnal cycles of simplified Wet Bulb Globe Temperatures (sWBGT, **a**–**c**) and heavy labor lost (**d**–**f**) for Doha, Qatar in August (**a**, **d**), New Delhi, India in July (**b**, **e**), and Atlanta, United States in July (**c**, **f**). The diurnal cycle is calculated from 2-m air temperature, surface pressure, and 2-m dew point temperature from hourly European Centre for Medium-Range Weather Forecast 5th Generation Reanalysis (ERA5) data. The bottom line shows the 2001–2020 mean, and the darker colored lines show sWBGT and labor losses for global-mean temperature changes (Global $\Delta T$) of +1 °C, +2 °C, +3 °C, +4 °C relative to the recent past. Local warming magnitudes are derived from Coupled Modeling Intercomparison Project, Phase 6 (CMIP6) warming projections ("Methods"). The reanalysis-derived climatological averages of sWBGT shown here are intended for illustrative purposes; hourly temperatures on a specific day can be higher than the 20-year mean diurnal cycles for these months.

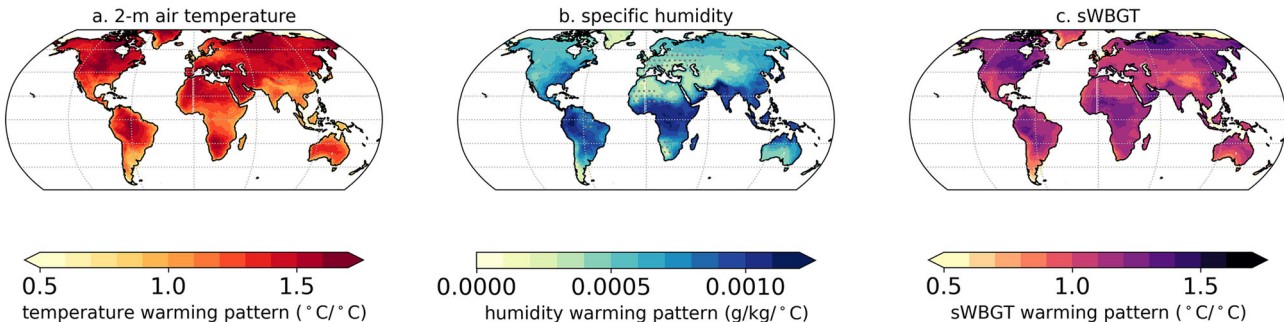

**Fig. 2 Warming patterns of temperature, humidity, and sWBGT in CMIP6 models.** Local warming patterns (local change per degree of global warming) for the annual 75th percentile of **a** daily mean 2-m surface air temperature, **b** daily near-surface specific humidity, and **c** daily simplified Wet Bulb Globe Temperatures (sWBGT) in Coupled Modeling Intercomparison Project, Phase 6 (CMIP6) 1%$CO_2$ simulations ($n = 21$). Maps show multi-model median values for each grid point, with stippling on maps where there is disagreement in the magnitude of local change (coefficient of variation >0.35; "Methods").

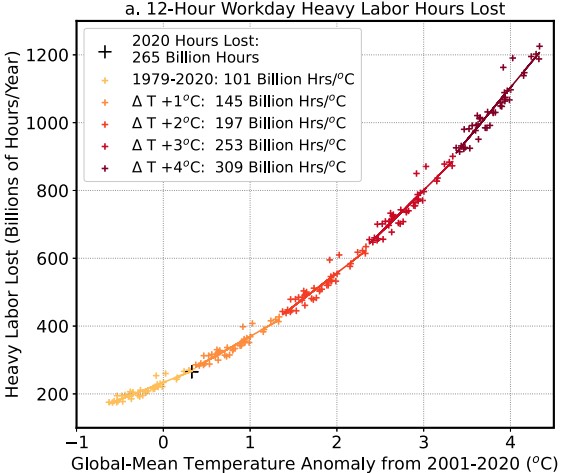
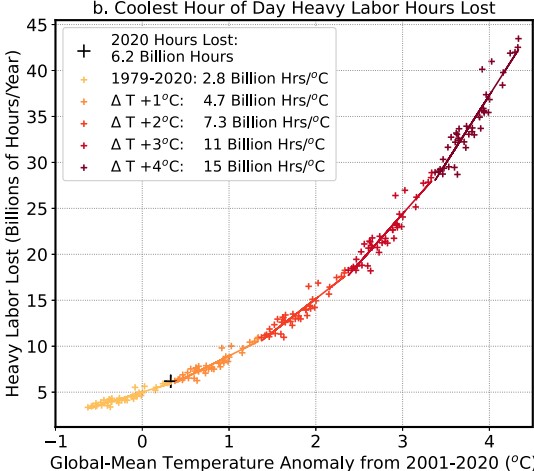

**Fig. 3 Relationship among global air temperature and labor losses.** Global-mean temperature anomalies (relative to the 2001–2020 mean) and global sums of heavy labor lost for the full 12-h workday (**a**) and for the coolest hour of the day (**b**) for 1979–2020 and with +1 °C, +2 °C, +3 °C, and +4 °C of additional global warming. Darker colors indicate higher warming levels. For future warming relationships, Coupled Modeling Intercomparison Project, Phase 6 (CMIP6) simplified Wet Bulb Globe Temperatures (sWBGT) warming patterns (Supplementary Fig. 5) are added to the 1979–2020 reanalysis-based sWBGT data for each global temperature change (+1 °C, +2 °C, +3 °C, +4 °C). Lines show best-fit regression lines for the given climate state (1979–2020, +1 °C, +2 °C, +3 °C, +4 °C), and the legend lists the slopes of these regression lines. Crosses mark global sums of heavy labor hours lost for individual years, with the year 2020 shown in a black cross.

patterns shown in Supplementary Fig. 5) to study heat exposure impacts in the present-day climate, and under a range of global-mean temperature changes (present-day, and with +1 °C, +2 °C, +3 °C, and +4 °C of global temperature changes relative to the recent past). In the present climate, productivity loss is concentrated in the tropics and subtropics, but as the planet warms, losses expand into the mid-latitudes, and losses in the tropics increase[28], both in the coolest hour of the day around sunrise and in the hours after sunrise (Supplementary Fig. 8).

We find a strong relationship among global-mean, annual-mean temperatures and global, annual sums of labor lost both in the recent past and with warming (Fig. 3). When we regress global temperatures against annual, global sums of labor lost from the ILO heavy labor sectors, we find that in the last 42 years (1979–2020), ~101 billion hours/year (±6 billion hours) of additional work was lost in the 12-h workday per degree of global warming ($r^2 = 0.89$). The relationship among global-mean temperature and global sums of labor loss at the coolest hour of the day is similarly strong (2.9 billion additional hours per degree of global warming, $r^2 = 0.88$).

To examine the shifting relationships among global temperatures and labor losses in a warmer climate, we add CMIP6 warming patterns (+1, +2, +3, +4 °C) to the ERA5 sWBGT (1979–2020) and regress global-mean air temperatures against labor losses under these warming levels ("Methods"). Air temperature is used in the calculation of sWBGT ("Methods"), and sWBGT increases relatively linearly as the globe warms[14,16,17], but the relationship among global temperatures and labor losses is nonlinear. As the globe warms, more land area is exposed to heat exposure that reaches unsafe levels for continuous work (Supplementary Fig. 8), and more hours of the day in warm locations become too hot for continuous labor (Fig. 1). Therefore, the relationship among global-mean temperatures and labor grows nonlinearly in the coming century. For example, the number of hours lost in the 12-h workday increases from ~101 billion hours/°C in the last 42 years to 197 billion hours/°C (±11 billion hours) with an additional 2 °C of global warming (Fig. 3). Similarly, productivity losses at the coolest hour of the day also increase non-linearly; losses at the coolest hour of the day increase from ~2.8 billion hours/°C (±0.2 billion hours) in

the last 42 years to 7.3 billion hours/°C (±0.4 billion hours) with 2 °C of global warming (Fig. 3).

**Warming impacts on local and global adaptation potential via time shifting.** Across the tropics and subtropics, much of the 12-h workday is already warm and humid enough to make large portions of the day unsafe for continual heavy labor (Fig. 1, Supplementary Fig. 8), so we also examine how much time could be recovered if workers could move heavy labor from the hottest hour of the day to the coolest hour or to an early morning hour (Supplementary Fig. 9). This analytical choice is motivated by a recent study that found Indonesian workers are already avoiding work in the peak heat of the day[23], as well as another study that recommends agricultural workers shift 1–2 h of work to the early morning to increase productivity[24]. We find that in the current climate, ~25–30 lost hours/person/year could be 'recovered' if workers in many low-latitude locations could move heavy labor to a cooler hour from the hottest hour of the day. In a warmer world, because midday temperatures remain hotter than the morning hours, adaptation becomes even more important. However, even though the early morning hours remain cooler than the midday hours, temperatures in the coolest hours of the day will increase as the planet warms (Fig. 1), and in some locations may warm faster than the daytime maximum temperatures (Supplementary Fig. 4). Therefore, we also examine 'adaptation potential', or the percent of work time that is recoverable by moving work to cooler hours.

As the globe warms, although the absolute number of hours that are recoverable via adaptation increases (left columns Supplementary Fig. 8), the percent of time that can be recovered decreases as early morning hours warm enough to create conditions that are unsafe for continuous labor (Fig. 1). Global sums of estimated midday heavy labor hours lost show that 79% of work time lost at the midday hour can be recovered in the present climate if workers could move labor from the hottest hour of the day to an early morning hour, and 83% of the midday hour can be recovered by moving labor to the coolest hour of the day (Fig. 4b). However, with 4 °C of additional global warming, global adaptation potential decreases to 65% via moving labor to the morning hour, and to 69% via moving labor to the coolest hour of

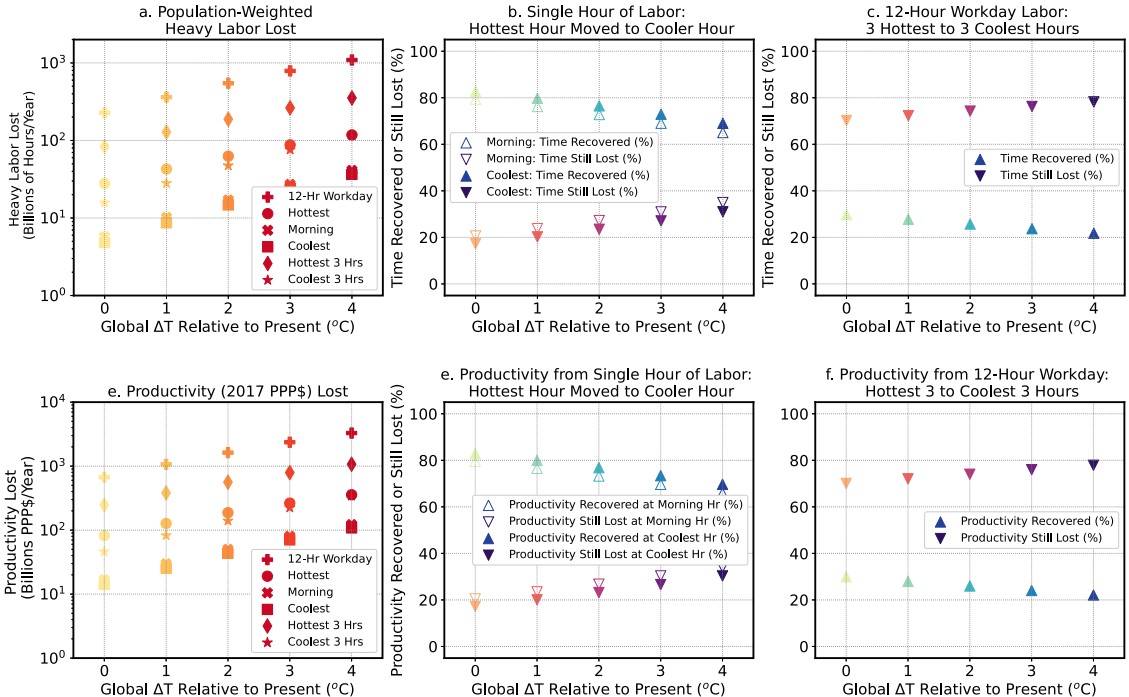

**Fig. 4 Global heavy labor and productivity losses and reduced global adaptation potential with warming.** Global sums of population-weighted heavy labor lost (**a**–**c**), and productivity (purchasing power parity-adjusted international dollars, or 2017 PPP$) lost (**d**–**f**) for global-mean temperature changes (Global $\Delta T$) relative to the recent past (2001–2020) of +1 °C, +2 °C, +3 °C, and +4 °C. Global sums of heavy labor lost (**a**) and productivity lost (**d**) weighted by working-age population in heavy labor ("Methods") for the full 12-h workday, the hottest hour of the day, a morning hour (the third coolest hour of the day), the coolest hour of the day, the 3 hottest hours of the day, and the 3 coolest hours of the day. Percent of time (**b**) or productivity (**e**) that can be recovered (or is still lost) by moving labor from the hottest single hour of the day to the morning hour (unfilled triangles) or to the coolest hour of the day around sunrise (filled triangles). Percent of lost work time (**c**) or productivity lost (**f**) in the 12-h workday that can be recovered if work in the 3 hottest hours of the day is replaced with work in the 3 coolest hours of the day. Note the log y-axis scale for the left panels showing global sums of labor or productivity lost. Productivity is shown in units of 2017 PPP$ equivalent ("Methods"; Supplementary Text 1).

the day. Notably, the rate of loss of adaptation potential (i.e., the rate of loss of work time in the coolest hour relative to the hottest hour) increases from <3%/°C of global warming to 3–4%/°C if the globe is 3–4 °C warmer than present (Fig. 4b).

Although moving a single hour of labor from the hottest hour of the day could allow workers to recover productivity lost during this hour[24], the 12-h workday is often the focus of global labor assessments[11,13]. Therefore, to provide perspective on how much labor is recoverable by moving several hours of work[26], we also calculate the percent of labor lost during the 12-h workday that is recoverable by replacing work conducted during the hottest 3 h with work conducted during the coolest 3 h of the day, assuming daylight is not a limiting factor. Currently, on a global scale, 30% of the labor lost during the 12-h workday can be recovered by moving these three hours. However, global workshift adaptation potential decreases at a rate of ~2%/°C of warming. Therefore, if the globe warms an additional 4 °C, only 22% of the workday can be recovered by moving labor from the hottest hours of the day (Fig. 4c). In some industries (e.g., construction or resource extraction) in specific locations, laborers are already able to work at night, so we also assess the ability of workers to recover lost labor by moving work from the hottest 12 h of the day to the coolest 12 h of the day. We find that currently, on a global scale, 62% of the labor lost during the 12-h workday can be recovered by moving the entire shift to the coolest 12 h of the day. However, global 12-h workshift adaptation potential decreases at a rate of about 3–4%/°C of global warming. Notably, under an additional 2 °C of future warming, more global labor would be lost in the coolest 12 h of the day than is currently lost in the hottest 12 h of the day (Supplementary Fig. 10).

The current and projected economic costs of heavy labor losses are substantial[28]. We find that already each year sees nearly 670 billion purchasing power parity-adjusted international dollars (2017 PPP$) lost globally in the 12-h workday, with 82 billion 2017 PPP$ lost at the hottest hour of the day and 14 billion 2017 PPP$ lost in the coolest hour of the day ("Methods"). Economic productivity losses associated with heat exposure accelerate as the globe warms, with 1.6 trillion 2017 PPP$ annual losses in the 12-h workday in a 2 °C warmer world (Fig. 4d). If workers attempt to minimize productivity losses by shifting work hours from midday to the early morning, losses in the morning relative to those at midday increase from about 2–3%/°C for the next degree of warming, to 3–4%/°C of warming under higher warming levels (Fig. 4e, f).

## Discussion
In the face of a warming world, workers are already shifting schedules to limit midday heat exposure[20–23], but daylight hours are limited[26], and as shown here, background global warming will increasingly restrict the ability of workers to adapt to warming by time shifting (Fig. 1). Even at the coolest hour of the day, there are currently several billion hours of heavy labor lost per year globally (Supplementary Fig. 2), with labor losses in the early morning hours increasing nonlinearly as the globe warms (Fig. 3). The relationship among global temperatures and global total labor losses and economic productivity losses is inherently non-linear as the background climate state changes and the geographic extent of heat exposure increases (Figs. 3, 4; Supplementary Fig. 8).

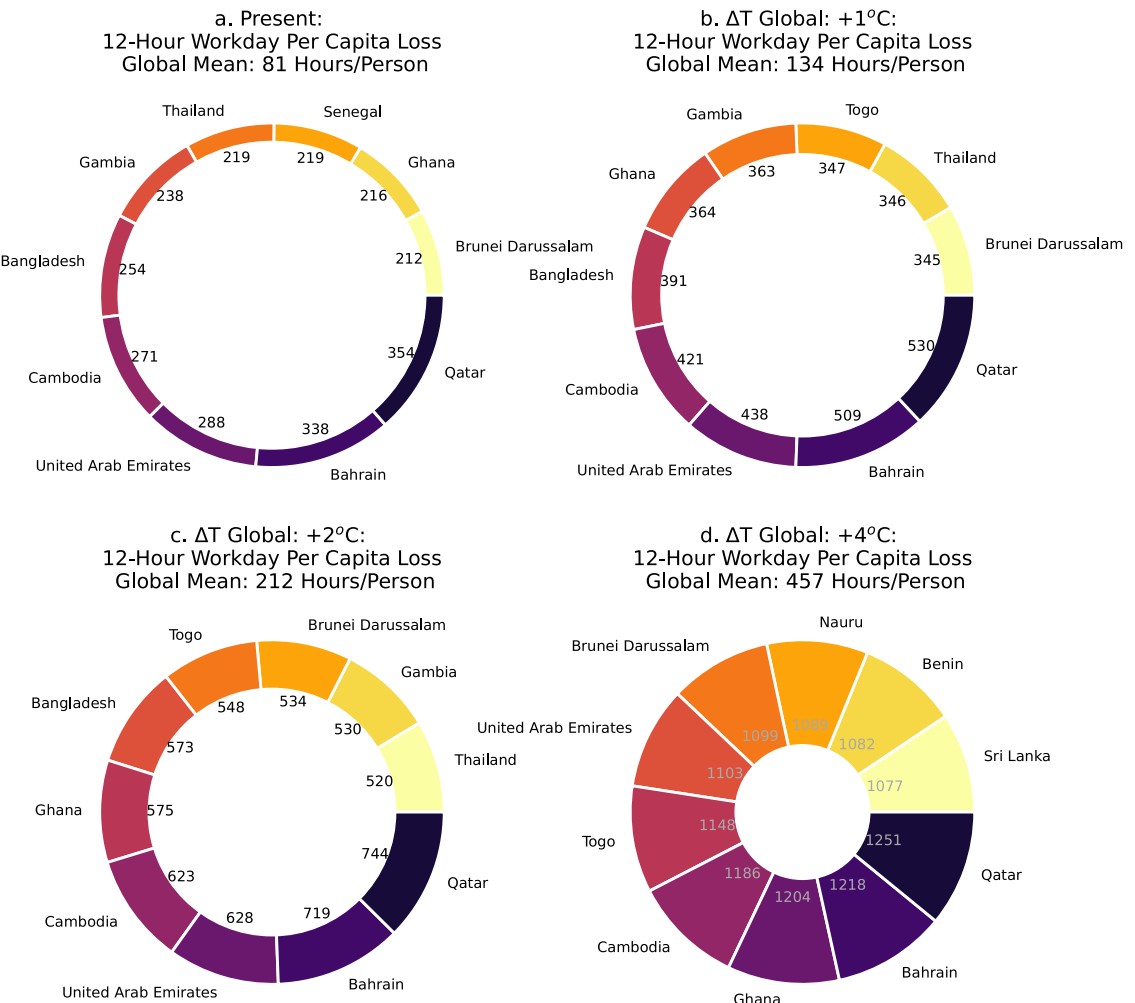

**Fig. 5 Countries with the highest per-capita labor losses from heat exposure.** The ten countries with the most per-capita labor losses in the 12-h workday in heavy labor due to heat exposure in the present (**a** 2001-2020 mean), and with +1 °C, +2 °C, and +4 °C of additional global warming (**b–d**, Global Δ*T*). The global mean of labor lost (average of 163 countries with available data; Supplementary Table 1) is shown above each plot, the numbers around the center of the circle show individual losses per country, and the thickness of the circle increases as the global mean of labor lost increases with warming. All units are in hours/person/year.

Labor losses from heat exposure are spatially variable, with several countries in Southwest Asia, South Asia, and Africa that already experience per-capita, 12-h workday labor losses > 200 h/person/year. Qatar and Bahrain show the worst impacts, with >300 h/person/year labor losses (Fig. 5a). In the coolest hour of the day, Qatar and Bahrain are still the most impacted by heat exposure (15 h/person/year lost), followed by several island and coastal nations in the western Pacific, which show losses >10 h/person/year at this coolest hour (Supplementary Fig. 11). When we overlay per-capita labor losses on the working-age population in heavy outdoor labor (Methods), we find that countries with large populations in South and East Asia experience the most work hours lost, both in the coolest hours (not shown) and in the full workday (Fig. 6a), with India showing the largest heat exposure impacts on heavy labor (>101 billion hours lost/year), despite its modest average per-capita labor losses (162 lost hours/person/year). Large population-weighted labor losses (>10 billion hours/year) in other countries such as Pakistan, Bangladesh, and China are driven by a combination of large working-age populations, seasonal heat exposure, and large fractions of the population that work in agriculture and construction industries (Supplementary Fig. 12). Under future warming, India, China, Pakistan, and Indonesia experience the largest population-weighted labor losses (Fig. 6b–d) and associated

economic productivity impacts (Supplementary Fig. 13), despite having lower national-average per-capita losses than other countries with smaller populations in Southeast Asia and tropical Africa (Fig. 5b–d). Bangladesh is a notable exception as it shows large per-capita as well as population-weighted labor losses currently and with warming.

Our accounting assumes that individuals are losing work productivity in the heat. Indeed, laborers who are encouraged to self-pace may regulate their own workload to maintain comfort[19]. However, worker productivity is linked to economic incentives, which is in turn linked to the health and well-being of workers, so individuals may continue to work at the detriment to their health, such as when they are paid by the piece for work[29–31]. If laborers are unable to work under safe conditions, they are at higher risk of multiple health impacts, including premature death[32–35], workplace injuries[36], morbidity from heat-related illness[37,38], traumatic injuries[7,39], and acute kidney injury[31]. Heat exposure is also implicated as a potential contributing factor to an epidemic of chronic kidney disease of unknown etiology in otherwise healthy, relatively young workers in Central America, Sri Lanka, India, and Egypt, and other areas[40,41]. Heat exposure can also increase the absorption of certain chemicals[42] and is associated with adverse pregnancy[43] and mental health outcomes[44].

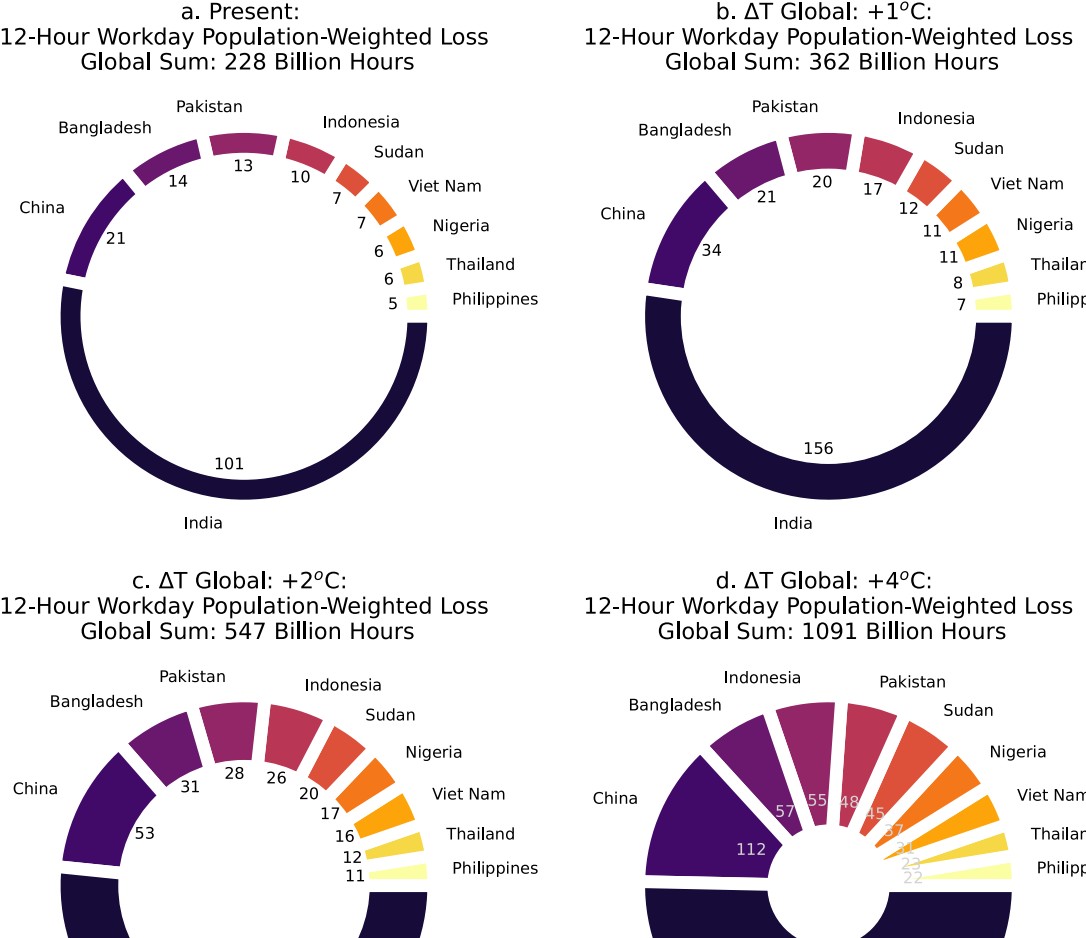

**Fig. 6 Countries with the most population-weighted labor losses from heat exposure.** The ten countries with the most heavy labor losses in the 12-h workday due to heat exposure weighted by the working-age population in outdoor heavy labor industries in the present (**a** 2001–2020 mean), and with +1 °C, +2 °C, and +4 °C of additional global warming (**b–d**, Global ∆T). The global sum of labor lost (sum of 163 countries with available data; Supplementary Table 1) is shown above each plot, the numbers around the center of the circle show individual losses per country, and the thickness of the circle increases as the global sum of labor lost increases with warming (details in "Methods"). All units are in billions of hours/year.

We have focused on worker adaptation via moving work hours from the middle of the day to the early morning hours[23,24,26]; governments have already implemented mandatory work breaks during the hottest parts of the day in locations such as in the United Arab Emirates[19]. However, implementation of this approach is situationally dependent and can result in unintended consequences. First, moving work to earlier hours may impact sleep duration, which is associated with injury risk[45,46]. Furthermore, heat exposure can affect sleep[47], which can affect the risk of injury and heat strain. Approaches to optimize sleep hygiene, and consideration of impacts on circadian rhythms and sleep, should be included in plans to shift work hours. Second, occupations and industries (e.g. construction) in certain settings may be limited in their abilities to shift work hours due to policies such as local noise ordinances[48]. However, policies that restrict night shift or early morning work, such as noise restrictions, may not be permanent barriers to adaptation if future investigation of potential adaptation strategies prompt changes in local ordinances to accommodate these strategies. Also, changing work

hours has the potential to introduce additional hazards related to other aspects of ambient conditions, such as lighting. These factors should be anticipated and addressed when optimizing work hour timing. Finally, changes in work schedules need to be coordinated with childcare and other obligations to maintain overall community well-being. Workers and communities should be included in decision-making to ensure that important considerations are not overlooked. Nonetheless, our findings provide baseline climate information about shifting work times, which is critical to informed decision-making about the most promising combination of approaches at different levels (e.g., individual, workplace, community, policy).

Future work should also consider including other heat stress metrics[17], because different metrics are best used for approximating heat stress in distinct situations, such as considering extent of perspiration and clothing[49]. We have also used reanalysis-based hourly estimates of heat exposure, but this method is known to be conservative as it underestimates extremes observed at weather stations[1]. Additionally, to better account for

uncertainty in projections of climate impacts, future approaches could incorporate more detailed on-the-ground data related to environmental conditions, work-rest cycles, work pace, work organization, physiological heat strain[50], and other ERFs that relate work capacity to heat exposure[51]. Although reanalysis-based global estimates of annual heavy labor losses due to heat exposure approach several hundred billion hours per year[13] (Supplementary Fig. 2), there are relatively few field-based studies that quantify work time lost due to heat exposure[5,23]; more field observations are needed to verify the results presented here. Additionally, this study has focused on one specific adaptation mechanism, shifting of work to cooler hours. Future work could consider other adaptation strategies, such as task shifting (e.g., movement of labor-intensive tasks to cooler hours), the limits of time and task-shifting strategies, what these mechanisms will cost both together and separately in terms of lost productivity, as well as recommendations for how workers could make choices between adaptation options and weigh their utility under various warming levels and under different scenarios.

With global-mean temperatures now over 1 °C warmer than a century ago, the Earth's climate is still within a regime that makes moving worker hours an approach – when combined with other adaptation mechanisms – that can help cope with warming temperatures. If future warming can be limited, this time-shifting adaptation mechanism may remain an effective option for many locations in the tropics and subtropics (Fig. 3, Fig. 4). However, with additional warming, this adaptation mechanism becomes less efficient as unsafe heat exposure in the morning hours magnifies in the tropics and subtropics, and expands into the extratropics. An additional 1 °C of global warming relative to the present could occur as early as 2037, and another 2 °C of warming could occur as early as 2051 (Supplementary Fig. 14). Therefore, if warming is left unchecked, the globe will continue to move into a new, 'less adaptable' climate regime within the lifetime of many young and middle-aged workers. These results further highlight the need to find alternative adaptation mechanisms to keep workers safe as well as to limit future warming to 1.5 to 2 °C[52] to help protect the livelihoods and health and well-being of workers in the low and mid-latitudes.

## Methods

**Heat exposure, labor losses, and associated economic costs**. There are a variety of methods for estimating heat exposure[17,53,54]. Wet Bulb Globe Temperature (WBGT)[55] is an internationally recognized heat stress metric that incorporates temperature and humidity measures and is used in occupational health studies and in military applications[56–59]. The value of the WBGT metric (or similar metrics that account for heat and humidity) is that it enables us to determine heat stress for both dry and humid heat in a common way (i.e., how easily the human body is able to cool itself). The ability to compare risk in both dry and humid conditions is essential in adaptation planning because use of air temperatures alone would not take into account the differences in heat stress due to variations in relative humidity throughout the day, or across seasons or locations. However, WBGT has its own limitations[60], and in some cases is not the best metric for measuring heat stress[61–63]. Additionally, WBGT can be difficult to measure and calculate as it requires specialized equipment that is not typically used at weather stations[56], and the necessary measurements needed to estimate WBGT are not output as variables from state-of-the-art model projections. Therefore, we focus on sWBGT to estimate heat exposure and labor impacts. sWBGT approximates WBGT using estimates of near-surface temperature, humidity, and pressure. The sWBGT metric used here assumes no solar radiation and is intended to estimate heat exposure in the shade or indoors with no air conditioning[1]. It is important to note that WBGT in the sun can be at least 2–3 °C higher than shade values[64], and reanalysis-based estimates of WBGT can underestimate extremes[1], so our estimates of productivity losses from heat exposure may be conservative. Nonetheless, these shade values of WBGT are used to estimate warming impacts on labor in the ILO report on labor on a warmer planet[11], in the Lancet Countdown on Health and Climate Change[13,65,66], and in other recent work[1]. Further details about calculation of sWBGT in reanalysis data can be found in the sections below.

We use an exposure response relationship based on epidemiological data to derive 'work ability' (WA) for a given hourly value of WBGT. This method[13,65,66] employs estimated exposure response relationships for reduced hourly work

capacity (labor productivity) for heavy manual labor conducted at 400 W intensity using hourly sWBGT and a cumulative normal (ERF) function:

$$\text{Loss fraction} = 1/2 \times (1 + \text{ERF}\,((\text{sWBGT} - \text{WBGTaver})/\text{WBGTSD} \times \sqrt{2})) \quad (1)$$

where for heavy work WBGTaver = 32.47 and WBGTSD = 4.16 for productivity lost per person per hour. Previous work[25,65,66] has assumed labor loss cutoffs of 10% and 90% of the hour, but here we use the approach employed in the 2020 Lancet Countdown on Health and Climate Change[13] that assumes the amount of work loss is defined by the exposure function. Although heat exposure can impact workers conducting light (e.g., services) and medium (e.g., manufacturing) labor, here we just consider heavy labor impacts because heavy labor losses account for the largest fraction of labor loss due to heat exposure[11,13,66].

We examine work losses in the 12-h workday, in the hottest hour of the day, in the third coolest hour of the day (here referred to as 'morning hour'), and in the coolest hour of the day (typically close to sunrise around 5–7 a.m. in the warm season). We also investigate the amount of time that could be recovered by moving labor from the three hottest hours of the day to the three coolest hours of the day, assuming daylight is not a limiting factor. To estimate labor losses during the 12-h workday, we calculate the daily mean sWBGT, daily maximum sWBGT, and the halfway point between these two values[25], and assume 4 h is spent near each of these values in the 12-h workday (4 × sWBGT max + 4 × sWBGT mean + 4 × sWBGT half). Although hourly weather reanalysis data are now available to calculate hourly losses in the 12-h workday, we have chosen to use the established '4 + 4 + 4' method due to computation and data storage constraints and for better comparison with previously published results[13]; further discussion of this method can be found in Supplementary Text 1.

The theoretical annual maximum work loss per person in the 12-h workday is 4380 h/year (12 h/day, 365 days/year), and for the hottest hour, morning hour, and coolest hour, is 365 h/year (or 1 h/day, 365 days/year). Here we focus on the 12-h workday based on the idea that most workers conduct their work between approximately 7 a.m. and 7 p.m.[13,25,28,66]. However, this method does not account for unsafe heat exposure outside of the traditional daylight work hours, such as late evening hours, when heat exposure is still high in many low-latitude locations (Fig. 1); during these 'non-traditional' work hours, unpaid household labor is often still conducted[67].

Our calculations likely provide conservative estimates of heat exposure for several reasons. Newly released, empirically based physical work capacity estimates[51] indicate that current methods[11,13] could underestimate work loss under heat and humidity levels previously thought to cause little to no productivity losses. Also, here we consider sWBGT calculated from ERA5 data, which, as previously mentioned, underestimates heat exposure extremes observed at weather stations[1]. In a location like the southeastern United States, sWBGT shows minimal to no labor productivity losses in the current average summer month (Fig. 1), but agricultural workers are already experiencing adverse heat health outcomes in many parts of the USA, so our estimates of labor losses underestimate actual worker risk[21,32,68,69]. The sWBGT metric used here considers heat exposure in the shade, so it will underestimate heat exposure in the full sun, and some work cannot be conducted in or moved to shaded areas. Finally, we do not take into account additional factors that could influence worker safety and productivity in the face of high heat and humidity such as ability of workers to use different clothing[69], underlying health conditions, varying degrees of acclimatization to heat, or hydration, among other factors.

We use ILO[70] estimates of numbers of workers in each country who work in heavy labor, here defined as agriculture+forestry+fishing and construction[13] to quantify the number of heavy labor work hours lost. For each of the countries in the dataset that have relevant ILO data that overlapped with the World Bank data (n = 163), we use the fraction of the overall working-age population (ages 15–64) in that country that works in heavy labor, multiply this fraction by the spatially gridded population ages 15–64 (Gridded Population of the World v4 data[71]), and then overlay the hours lost on the population data[13,25,65,66]. This method assumes that outdoor workers are geographically distributed similarly to the overall population, even if this is not always the case[69]. Nonetheless, for most countries sub-national information on work is not available, so we follow established methodology that distributes laborers with the general population.

We also estimate economic productivity decreases associated with lost earnings from heavy labor productivity loss. There are several methods to estimate economic costs, including multiplying hours lost by estimates of hourly earnings[13,72] and converting hours lost to job loss equivalents and associated productivity losses in terms of reduced contributions to Gross Domestic Product (GDP)[11]. We use the most recent World Bank GDP data to convert average productivity per worker in agriculture, forestry, fisheries and industry (including construction) to hourly output by assuming a 12-h workday, 365 days/year to maintain consistency with our hours lost estimates. We then multiply the hourly productivity per worker by the heavy labor hours lost to estimate economic costs of productivity losses due to heat exposure (details in Supplementary Text 1).

For future warming impacts on labor, we assume future population and earnings are static—in other words, they are fixed at levels and rates from the present. This is a conservative assumption for projecting future population impacts because future population is expected to rise, particularly in many low-latitude countries where heat exposure is projected to increase[73,74].

**Heat exposure metric calculation from reanalysis data**. Following the method of Li et al.[1], we calculate the sWBGT from hourly, single level (near-surface) ERA5[75] atmospheric reanalysis output (Jan 1, 1979 to Dec 31, 2020) using 2-m air temperature (t2m), surface pressure (sp), and 2-m dew point temperature (d2m) using the equation:

$$sWBGT = 0.7Tw + 0.3Ta \tag{2}$$

where Tw is 'isobaric wet bulb temperature' and Ta is dry air temperature (t2m). *Tw* is calculated from air temperature, dew point temperature, and surface pressure. ERA5 is provided at ~35 km spatial resolution at the equator, and climate model data are regridded to this grid resolution via bilinear interpolation when the model-based warming patterns are added to the reanalysis data. When ERA5 sWBGT and hours lost data are compared to population data, the hours lost spatial data are regridded to the population resolution (~0.5 × 0.5 degree spatial resolution). We overlay a spatial mask of each country's borders to calculate the country-by-country labor loss estimates by sector and worker productivity losses, and sum labor and productivity losses within the country borders to calculate country-level losses.

**Heat exposure metric calculation from climate model data**. Following the methods of[14], we calculate WBGT using 2-m air temperature ('tas' variable), near-surface specific humidity ('huss' variable), sea-level pressure ('psl' variable), and orography ('orog' variable) from CMIP6 models that provide the necessary variables (Supplementary Table 2) using the equation:

$$sWBGT = 0.567T + 0.393VP + 3.94[^\circ C] \tag{3}$$

where *T* is daily mean 2-m air temperature ('tas') and *VP* is vapor pressure. Vapor pressure (*VP*) is calculated from daily mean specific humidity ('huss'), sea-level pressure ('psl'), and orography ('orog'). We use the output from idealized 1%CO$_2$ simulations, which are only forced by increases in atmospheric CO$_2$ concentrations, starting at pre-industrial levels (~284 ppm), and increasing at 1% per year for 150 years[27]. Here we use model years 35–150 from the 1%CO$_2$ experiment because year 35 approximately coincides with present atmospheric CO$_2$ concentrations (~400 ppm). We choose the 1%CO$_2$ experiment because the 21st century Shared Socioeconomic Pathways (SSPs) include highly uncertain, theoretical future transient aerosol, land use, and other forcing changes[73]. To determine if warming patterns among experiments are robust, we compare warming patterns from 1%CO$_2$ to patterns from the SSP5-8.5[76] and find that warming patterns are similar (<10% difference in local magnitude), except in isolated locations in the mid-latitude northern hemisphere (see Supplementary Fig. 15 and Supplementary Text 1).

**Warming patterns in CMIP6 models**. We first calculate daily sWBGT for each CMIP6 model then calculate the monthly mean sWBGT for each grid point. We regress global-mean, annual-mean, latitude-area weighted 2-m air temperature ('tas') against monthly local sWBGT after smoothing global and local data using a 20-year lowpass filter. We use the multi-model median (MMM) regression coefficient from this calculation (e.g., local change in each month per degree of global change) as the 'pattern scaling' variable for different global-mean temperature changes examined here. We have also calculated the warming patterns for annual, JJA/DJF, the 75th percentile, the 95th percentile, and the 99th percentile of daily sWBGT and find minimal differences in MMM warming patterns for land regions between ~40° N and 40° S (Supplementary Fig. 6). Warming patterns are calculated on the model's native grid resolution, then spatially regridded using bilinear interpolation to the ERA5 resolution for calculation of MMM and for adding warming patterns to the ERA5 data. We choose the MMM instead of multi-model mean because local warming patterns among CMIP6 models are not normally distributed at some locations. Supplementary Fig. 3 shows sWBGT warming patterns from individual CMIP6 models. To show where CMIP6 models agree on the magnitude of local change per degree of global warming, we calculate the coefficient of variation (inter-model local standard deviation divided by the local MMM). We stipple locations on maps (Fig. 2, Supplementary Figs. 4–7) where the coefficient of variation is > 0.35 to show where models do *not* indicate there is good agreement on the magnitude of local change relative to the MMM[16].

**Applying warming patterns to reanalysis data to estimate future heat exposure**. Here we are interested in quantifying heavy labor losses due to heat exposure, with a focus on heat exposure in the peak heat of the day (daily maximum), in the morning, and in the coolest hour of the day (typically around sunrise). We combine monthly sWBGT warming patterns from CMIP6 models with hourly sWBGT ERA5 data to estimate future heat exposure impacts on labor. This combination allows us to rely on the instrumental-based background climate mean state and available model data without the need to bias correct model data. For the present-day climate, we calculate hourly sWBGT in ERA5 data, apply the productivity loss equation, and then calculate the mean work hours lost for the time period 2001-2020. To calculate future productivity losses, we add the monthly warming patterns from CMIP6 models to the hourly ERA5 sWBGT data (e.g., for a global warming of 2 °C in January, we multiply 2 by the local January CMIP6 warming pattern, then add this number to the hourly January ERA5 sWBGT data). We use monthly warming patterns because warming patterns can vary by season (Supplementary Fig. 5). We conducted a sensitivity test using warming patterns from the 75th

percentile of sWBGT and hemispheric summer warming patterns (JJA and DJF), and our main results did not change (not shown). After adding CMIP6 sWBGT warming patterns to the hourly sWBGT ERA5 data, we then calculate hourly labor lost in the 'pattern scaled' ERA5 data and average the labor lost over this 20-year time period to estimate the mean labor lost in a warmer climate. We assume all parts of the sWBGT diurnal cycle will shift equally. We make this assumption because models do not generally agree on the sign of difference in future changes in the daily maximum vs minimum temperatures, and differences in the magnitude of change in maximum vs minimum are <25%, or <0.2 °C per degree of global warming (Supplementary Fig. 4; Supplementary Text 1).

**Reporting summary**. Further information on research design is available in the Nature Research Reporting Summary linked to this article.

## Data availability
CMIP6[27] experimental output ('orog' and daily data variables: 'tas', 'tasmax', 'tasmean', 'huss', 'psl') can be found at: https://esgf-node.llnl.gov/search/cmip6/. Hourly single level reanalysis ERA5[75] data can be found at https://cds.climate.copernicus.eu/#!/search?text=ERA5&type=dataset. ILO sector-specific labor and earnings data can be found on the ILOSTAT data explorer[70]: https://www.ilo.org/shinyapps/bulkexplorer7/?lang=en&segment=indicator&id=SDG_0111_SEX_AGE_RT_A. World Bank data can be downloaded from: https://data.worldbank.org/. GPW v4 population data[71] are available at: https://sedac.ciesin.columbia.edu/data/collection/gpw-v4. The data generated in this study (average diurnal cycles of sWBGT, gridded CMIP6 monthly warming patterns, gridded heavy labor productivity losses, and global sums of heavy labor losses) have been deposited in the Zenodo database https://doi.org/10.5281/zenodo.5594470.

## Code availability
Python code provided by Li et al.[1] to calculate hourly sWBGT from ERA5 data are available on GitHub (https://github.com/dw-li/WBGT). Python code were provided by Chavaillaz et al.[14] to calculate sWBGT from CMIP data. Code is available from Chavaillaz et al., or the corresponding author, upon reasonable request. Python code and packages used to plot diurnal cycles of sWBGT, CMIP6 monthly warming patterns, and global sums of labor losses are available on GitHub: https://github.com/LukeAParsons/Warming_Adaptation_Limits.

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

## Acknowledgements
L.A.P., Y.Z., and D.S. acknowledge NASA GISS grant number 80NSSC19M0138 for funding support. Y.Z. acknowledges NASA Health and Air Quality Applied Sciences Team grant number NNX16AQ30G. We acknowledge the World Climate Research Program, which, through its Working Group on Coupled Modeling, coordinated and promoted CMIP6. We thank the climate modeling groups for producing and making available their model output, the Earth System Grid Federation (ESGF) for archiving the data and providing access, and the multiple funding agencies who support CMIP6 and ESGF. G. Faluvegi, M. K. Brennan, and P. Goddard provided helpful feedback about analytical methods and coding. For CMIP the U.S. Department of Energy's Program for Climate Model Diagnosis and Intercomparison provides coordinating support and led development of software infrastructure in partnership with the Global Organization for Earth System Science Portals. This work contains modified Copernicus Climate Change Service Information (ERA5 data).

## Author contributions
L.A.P., D.S., and M.T. planned the analysis. L.A.P. and Y.Z. collected the data. L.A.P. conducted the analysis. L.A.P., D.S., M.T., Y.Z., and J.T.S. contributed to writing and editing of the manuscript and to the interpretation of the data.

## Competing interests
The authors declare no competing interests.
