## [Peer Review File · Nature Communications]

Reviewer comments, first round review –

Reviewer #1 (Remarks to the Author):

This was an excellent body of work that takes worker heat exposure modelling to the next level. Quite complex methodology and data analysis. Conclusions have of course have limitations as would be expected by using surrogate measures to calculate WBGT. The most significant of this being the fact that globe temperature cannot be modelled and many workers are in fact exposed to direct sunlight. The paper does highlight this limitation.

One issue that might be considered in future work may be to use an alternative heat stress index, WBGT work-rest regime can be a bit over protective. In hot climates I do believe acclimatisation is a factor that leads to self pacing of work and many "hot" tasks are already scheduled for cooler parts of the day.

Congratulations on a very worthwhile body of work that draws attention to the cost of heat impacts.

Reviewer #2 (Remarks to the Author):

This paper builds on the latest models and projections of extreme heat under various climate change scenarios. It offers an essential analysis that goes beyond the extant literature which estimates of labour loss without adaptive measures, to offer an analysis of the effectiveness and limits of a particular adaption measure - that of rescheduling intensive or exposed work to cooler times of the day. This is an extremely valuable paper in that it enables basic estimates of the value (lost and gained) of this adaptation strategy in relation to hours of productivity and economic value. The paper builds on widely accepted metrics and methods, meaning that the findings are easily translatable and of immediate use to these policy arenas.

The paper has a few areas where it can be a little clearer or developed further to make its significance - which is substantial - more readily apparent, namely in a) how it understands extreme heat and heat stress and relates this to (s)WBGT and climatic conditions; b) the precise nature of the adaptation strategy it focuses on, and c) the implications of the limits it identifies in using this strategy - as part of loss and damage calculations and thereby as a platform for alternative adaptation options - and to support more informed and detailed decision-making about the portfolio and timing of adaptation strategies for countries, sectors and companies/organisations. d) where next - how this analysis can be used in more detailed scenario exercises and adaptation planning. [These points are detailed below.]

I have restricted the focus of my comments to the occupational health and productivity and climate adaptation framing and policy implications of this paper, as these relate to my fields of expertise. As non climate-scientist/lay reader of these aspects of the paper, I will note that it was one of the more accessible methods sections for such work that I have read, which I appreciated and think is valuable for the wider academic community.

Detailed comments:

a) The way the term 'humid heat' is used in the paper and its relationship to WBGT (and sWBGT) are a little unclear to me - I have the impression that (s)WBGT and humid heat are somewhat conflated/treated as synonyms and/or that WBGT is only of use for humid heat and not for dry heat.

For example, the commentary on Qatar suggests only humid heat presents a risk to workers, whereas it predominantly has hot and dry weather which also presents a risk. The value of WBGT (and similar) is that it enables us to interpret both dry and humid heat in a common way - i.e. in relation to the heat stress conditions (including low and high humidity) cause. This is the reason

why WBGT is useful for humid heat, as the heat stress this causes is typically under-represented by ambient Temperature, but it is also useful for more accurately assessing the heat stress caused by dry and hot conditions (which might otherwise be over-estimated).

As I understand it, because your paper assesses (s)WBGT, it is useful for understanding heat stress risk places that typically experience humid heat but also in places that experience dry heat, or both. The ability to compare risk in both dry and humid conditions is essential in adaptation planning either during the day or across season, for example in the mornings when temperatures are lower but WBGT could be higher - this is an additional reason why your analysis using sWBGT exposure and the benefit of work-time shifting is important, as assessments using ambient temperature only may have over-estimated the efficacy of this particular adaptation strategy.

Edits for clarity on the relationship between WBGT and humid heat would therefore present a more accurate picture of what your research does, and makes clearer its value for adaptation planning.

b) Time shift -vs- task shift. An edit for clarity and consistency throughout the paper on whether you mean shifting work hours (i.e. cessation of any kind of work/designation of non-working hours in the hottest day and moving those hours to a cooler period, changing the commencement and/or duration of the work day) OR shifting more exertional and/or exposed tasks from hotter hours to cooler hours, while retaining the original commencement and ending times of the shift would be helpful. Task-shifting/rescheduling might be seen as complementary to time-shifting, or there could be progression from one-to the other as climate change progresses. It would be great to spell out more clearly how your analysis supports weighing up these options.

c) Regarding the point above, although other heat management and adaptation strategies are mentioned, sometimes the paper seems to conflate time/task shifting with adaptation per se. Yes, overall adaptive capacity is limited by the limits to the efficacy of this particular option, but it would be good to see some more detailed examination of how that knowledge supports more robust adaptation planning - e.g.

-1: The limits of time/task shifting strategies and what it will cost in terms of lost productivity and GDP, calculated as stand-alone response. This could support arguments for funding alternative adaptation measures (e.g. through Warsaw International Mechanism for Loss and Damage).

- 2: choices between adaptation options, how to assemble a collection of strategies and weigh their utility including over time and under different scenarios (e.g. compensating for lack of value from time shifting by investing in active cooling interventions).

Minor comments -

Line 29 - the grammatical structure of the sentence implies avoiding unsafe working conditions *cause* labour productivity losses directly - whereas the latter usually refers only to a reduction in work rate, which may in fact be a sign of self-pacing and an appropriate response to heat. Suggest rephrasing (such as " ...unsafe and causes labour productivity losses ..').

Line 42 - Consider costs as well as benefits, to avoid bias to positive outcomes.

Paragraph from Line 180 - heat strain impairing physical and cognitive function and contributing to higher accident and workplace injury rates is also worth mentioning - e.g. the just published study from Park, Pankratz and Behrer (2021).

269 - A consideration of formal (night) shift work would be valuable to consider in relation to your findings (e.g. swapping the 12 hottest hours for the 12 coolest hours). Large construction sites and resource extraction and processing are examples of heavy labour contexts where night shifts are often used (although usually in the context of 24-hour operations).

From a policy perspective, it is helpful to indicate in the main body/introduction to the paper that analysis of impacts is conservative, given use of ERA5 and basing the analysis on shade conditions

underestimates actual WBGT.

An indication of future research, and whether your analysis and data are available for use would be helpful.

Similarly, policies that restrict night shift/early morning work such as noise restrictions, industrial zoning etc should not be assumed to be permanent barriers to adaptation, but as triggers for a more extensive investigation of potential adaptation strategies and a reimagining of what a heat-adaptive society might look like.

The paper already makes a strong and useful argument, and provides valuable findings. With a bit more clarity on the above points it would offer an even more effective launch-pad for further research and more detailed adaptation planning by relevant policy communities.

Reviewer #3 (Remarks to the Author):

In this manuscript, Parsons et al. develop estimates of global labor productivity loss resulting from humid heat conditions in the present day as well in the future with different amounts of human-induced warming. The authors then quantify the percentage of productivity that could be recovered under several work-shifting scenarios that model changes in when heavy labor is performed within the standard 12-hour workday they modeled.

The questions the authors are asking with this analysis—namely, how will climate change affect the labor productivity of outdoor workers around the world? And how will the capacity to adapt to warming change as that warming grows more severe—are interesting ones. And they are important questions to be considering as nations head into the COP26 international climate negotiations, at which discussions of and commitments to reducing emissions and paying for the costs of climate adaptation take place. Thus, this paper is an important contribution to our collective understanding of the costs of climate change.

Overall, this is a strong piece of research. The manuscript is well written, the methods the authors employed were sound—though there are some comments below that I'd like to see addressed-- and their conclusions followed reasonably from their results. Yet neither the methods nor the findings struck me as novel enough to warrant publication in *Nature Climate Change* given the caliber of the journal. For example, in performing their analysis, the authors essentially employed a previously published methodology for simulating futures with different amounts of warming relative the preindustrial era (i.e., those of Tigchelaar et al. 2020). And the core findings of increasingly severe heat constraints on labor are conceptually similar to the work of Dunne et al. 2013 and Kjellstrom et al. 2018, though it's notable that neither of those previous studies considered potential adaptation measures or adaptation capacity changes as the present manuscript does.

Comments:

1. In lines 55-57 and Figure 1, the authors describe locations where humid heat is already at or approaching levels unsafe for continuous heavy labor in the morning and at midday. In looking at the peak WBGT values in Figure 1, however, it's unclear what that unsafe level is and how it relates to the WBGT_{ave} presented in lines 245-246 of the methods section. From the methods section, I assumed that threshold would be 32.47C, but none of the locations in Figure 1 approach that value. I may be misunderstanding, but some clarity around what that threshold WBGT value is and how it relates to the equation in the methods section is needed. I suggest adding a sentence for clarity around lines 55-57.

2. The authors find that global labor losses due to extreme heat already more than 200 billion hours per year, with greater losses during anomalously or particularly hot years. Given that these findings are based on historical data, the manuscript would be strengthened significantly if they were vetted against an independent source of data. Are there any estimates—global or for any given nation—of actual labor productivity losses due to humid heat over the observational time period? As the authors note in the discussion, outdoor workers often choose to work during the

heat despite the health risks because they need the income. And I wonder if the losses calculated here for the historical period reflect what has actually transpired over that time period. Whether vetting the results with an additional data source is possible or not, I'd suggest future exploring in the text how well those historical results are capturing reality.

3. Lines 87-88: The authors find a strong relationship between temperature and labor loss, and while the relationship is very clean as presented in Figure 2, it is largely unsurprising given that the labor loss calculation is directly tied to WBGT as described in the methods. Perhaps some text in this section that more fully describes the importance of this finding would make the findings more compelling.

4. The analyses quantifying labor loss during the hottest and coolest hours of the day seem somewhat arbitrary. Is there evidence to suggest that workers or employers would deliberately shift heavy work in this manner rather than shifting a full workday to cooler hours? As it's currently presented, analyzing the potential benefit of shifting one hour of heavy labor to a cooler time of day seems more of a scientific exercise than a practical exploration of how work might actually shift in response to warming temperatures. The shifting of three hours, however, seems a much more likely occurrence. If there's evidence to suggest one-hour shifts in work time are taking place, that would be helpful to include in the section starting on line 166. If not however, I'd recommend trimming the paragraphs on the one-hour shifts and expanding the text on the three-hour shifts in work schedules.

5. Lines 257-261 (Methods): The authors assume that the 12-hour workday is evenly split, with four hours at the daily maximum WBGT, four hours at the daily mean WBGT, and four hours at the halfway point between the two. However, the data presented in Figure 1 seem to imply a different distribution of WBGT values over the course of a 12-hour workday. A reference is made to Kjellstrom et al. 2018, but it would be useful to include an explanation of whether or how the 12-hour WBGT data shown in Figure 1 supports this assumption.

6. Lines 328-331 (Methods): The explanation of why the 1%CO₂ experiment was preferable to the more traditional emissions pathways from CMIP6 was somewhat unclear. The latter is said to have the potential to "create localized differences in the magnitude of warming," yet the authors then state that what they're interested in is local temperature changes. If the inclusion of non-CO₂ greenhouse gases causes local temperature changes, would that not be important to include here? I'd like to see a more compelling or more clearly stated justification for using the 1%CO₂ experiment.

7. Much of the methods section (particularly lines 364-398 or so) describes the temperature and humid heat warming patterns from CMIP6 models. The results here are central to the study, as those warming patterns are applied to the reanalysis data in order to simulate future humid heat conditions, and while the choice to unpack them in the methods section is understandable, I'd suggest crafting a paragraph or two describing these findings at a high level and including that text in the main body of the paper.

8. Similar to my previous comment, the results described in the methods section seem overly detailed and like they are largely explaining what is presented in the supplementary figures. I'd suggest a) pulling some of this results-focused text from the present methods section into the supplementary information; and b) presenting a few concise metrics that describe these results in the methods section.

Reviewer comments are shown in bold font.

Author responses are shown italic font.

Line numbers are from accepted changes document.

REVIEWER COMMENTS

Reviewer #1 (Remarks to the Author):

This was an excellent body of work that takes worker heat exposure modelling to the next level. Quite complex methodology and data analysis. Conclusions have of course have limitations as would be expected by using surrogate measures to calculate WBGT. The most significant of this being the fact that globe temperature cannot be modelled and many workers are in fact exposed to direct sunlight. The paper does highlight this limitation.

One issue that might be considered in future work may be to use an alternative heat stress index, WBGT work-rest regime can be a bit over protective. In hot climates I do believe acclimatisation is a factor that leads to self pacing of work and many "hot" tasks are already scheduled for cooler parts of the day.

Congratulations on a very worthwhile body of work that draws attention to the cost of heat impacts.

We thank the reviewer for their time, supportive comments, and suggestions. Line numbers are from 'accepted changes' document.

We have made several changes to the text based on the reviewer's suggestions:

- 1) In the Discussion we now mention the need to use alternative heat indices in future work. (Line 255)*
- 2) In the Methods section we now mention work that has shown that WBGT is not the best humid heat metric in all situations/environments (e.g., the work by Dehghan et al., 2012 in the Persian Gulf; Bates and Schneider, 2008; Holmer, 2010) (Lines 297-300)*
- 3) In the Discussion, we now emphasize the idea that workers in some cases may already be self-pacing work (Lines 222-224) or moving labor intensive tasks to the cooler parts of the day, citing the work by Miller et al. that suggests workers are self-pacing in certain work environments in the UAE. (Lines 234-238)*
- 4) We agree that intensive/'hot' tasks are already being scheduled for cooler parts of the day in many situations. In addition to mentioning the UAE mandatory work breaks during the hottest parts of the day, we also highlight 'on the ground' research, such as Masuda et al. 2019 (GEC) that shows that workers are already adapting to warmer conditions by stopping work in the peak heat of the day. (Lines 194-195, 234-238)*

We would also like to note that although the studies mentioned above found that WBGT was found to be overprotective or conservative in some cases, here we are assuming no direct solar radiation, and we are using reanalysis data to calculate sWBGT. Both of these methodological choices produce lower estimated sWBGT values than are felt 'on the ground' (e.g., see Li et al., 2020 supplementary figures showing Chicago heat wave data from a weather station vs reanalysis data). We have mentioned both of these points in the Methods section.

Reviewer #2 (Remarks to the Author):

This paper builds on the latest models and projections of extreme heat under various climate change scenarios. It offers an essential analysis that goes beyond the extant literature which estimates of labour loss without adaptive measures, to offer an analysis of the effectiveness and limits of a particular adaption measure - that of rescheduling intensive or exposed work to cooler times of the day. This is an extremely valuable paper in that it enables basic estimates of the value (lost and gained) of this adaptation strategy in relation to hours of productivity and economic value. The paper builds on widely accepted metrics and methods, meaning that the findings are easily translatable and of immediate use to these policy arenas.

The paper has a few areas where it can be a little clearer or developed further to make its significance - which is substantial - more readily apparent, namely in

- a) how it understands extreme heat and heat stress and relates this to (s)WBGT and climatic conditions;**
- b) the precise nature of the adaptation strategy it focuses on, and**
- c) the implications of the limits it identifies in using this strategy - as part of loss and damage calculations and thereby as a platform for alternative adaptation options - and to support more informed and detailed decision-making about the portfolio and timing of adaptation strategies for countries, sectors and companies/organisations.**
- d) where next - how this analysis can be used in more detailed scenario exercises and adaptation planning. [These points are detailed below.]**

I have restricted the focus of my comments to the occupational health and productivity and climate adaptation framing and policy implications of this paper, as these relate to my fields of expertise. As non climate-scientist/lay reader of these aspects of the paper, I will note that it was one of the more accessible methods sections for such work that I have read, which I appreciated and think is valuable for the wider academic community.

We thank the reviewer for their time and comments. We specifically appreciate that the reviewer noticed our attention to detail and clarity in the Methods section. We have responded to each of the reviewer's points (a-d) below and made changes to the manuscript where appropriate. Specifically, we have changed the 'humid heat' terminology to 'heat exposure' or simply 'sWBGT' and explain in more detail the advantages of using a metric such as sWBGT to account for dry heat as well as heat + humidity. Line numbers are from 'accepted changes' document.

Detailed comments:

- a) The way the term 'humid heat' is used in the paper and its relationship to WBGT (and sWBGT) are a little unclear to me - I have the impression that**

(s)WBGT and humid heat are somewhat conflated/treated as synonyms and/or that WBGT is only of use for humid heat and not for dry heat.

For example, the commentary on Qatar suggests only humid heat presents a risk to workers, whereas it predominantly has hot and dry weather which also presents a risk. The value of WBGT (and similar) is that it enables us to interpret both dry and humid heat in a common way - i.e. in relation to the heat stress conditions (including low and high humidity) cause. This is the reason why WBGT is useful for humid heat, as the heat stress this causes is typically under-represented by ambient Temperature, but it is also useful for more accurately assessing the heat stress caused by dry and hot conditions (which might otherwise be over-estimated).

As I understand it, because your paper assesses (s)WBGT, it is useful for understanding heat stress risk places that typically experience humid heat but also in places that experience dry heat, or both. The ability to compare risk in both dry and humid conditions is essential in adaptation planning either during the day or across season, for example in the mornings when temperatures are lower but WBGT could be higher - this is an additional reason why your analysis using sWBGT exposure and the benefit of work-time shifting is important, as assessments using ambient temperature only may have over-estimated the efficacy of this particular adaptation strategy.

Edits for clarity on the relationship between WBGT and humid heat would therefore present a more accurate picture of what your research does, and makes clearer its value for adaptation planning.

We thank the reviewer for pointing this out- we now mention the advantage of this metric when we introduce why we use sWBGT (Lines 60-63).

'The advantage of sWBGT, and similar metrics that account for heat and humidity, is that it enables us to interpret both temperature and humidity in relation to heat exposure and heat stress for working individuals.'

Throughout the manuscript, we have also now changed the terminology we use to 'heat exposure', which we explain includes both temperature and humidity (Lines 62-66):

'The term 'heat exposure' here refers to conditions that are either simply too hot or are hot and humid enough to cause labor losses; these conditions can include high temperatures with moderate to low humidity, or moderate to high temperatures with high humidity, both of which would impact individuals conducting heavy labor.'

We also added a description of the advantages of using the sWBGT metric in the Methods (Lines 291-297):

'The value of the sWBGT metric (or similar metrics that account for heat and humidity) is that it enables us to determine heat stress for both dry and humid heat in a common way (i.e., how easily the human body is able to cool itself). The ability to compare risk in both dry and humid conditions is essential in adaptation planning because use of air temperatures alone would not

take into account the differences in heat stress due to variations in relative humidity throughout the day, or across seasons or locations. ‘

b) Time shift -vs- task shift. An edit for clarity and consistency throughout the paper on whether you mean shifting work hours (i.e. cessation of any kind of work/designation of non-working hours in the hottest day and moving those hours to a cooler period, changing the commencement and/or duration of the work day) OR shifting more exertional and/or exposed tasks from hotter hours to cooler hours, while retaining the original commencement and ending times of the shift would be helpful. Task-shifting/rescheduling might be seen as complementary to time-shifting, or there could be progression from one-to the other as climate change progresses. It would be great to spell out more clearly how your analysis supports weighing up these options.

We thank the reviewer for asking us to clarify- here we are only accounting for time shifting. As we state in the Introduction (lines 49-53):

‘Here we combine heat exposure estimates from reanalysis data with patterns of warming from the latest climate model projections to examine ‘adaptation capacity’, or what percent of work time is recoverable if laborers move work hours from the hottest hours of the day to the cooler morning hours, and how work loss and adaptation capacity change as the planet warms.’

and again in the Results of the paper (lines 152-153):

‘Therefore, we also examine ‘adaptation potential’, or the percent of work time that is unrecoverable by moving work to cooler hours.’

To clarify the analytical focus and highlight the potential of task-shifting, we also added a sentence to the Discussion that emphasizes the time shifting focus of the paper (lines 266-267):

‘Additionally, this study has focused on one specific adaptation mechanism, shifting of work to cooler hours.’

c) Regarding the point above, although other heat management and adaptation strategies are mentioned, sometimes the paper seems to conflate time/task shifting with adaptation per se.

In the Results, we state that in the context of our analysis, we are defining ‘adaptation potential’ as (Lines 152-153):

‘the percent of work time that is unrecoverable by moving work to cooler hours.’

Additionally, in the Introduction, we state that in the context of our analysis, we are defining adaptation potential as (Lines 47-53):

‘..the feasibility of moving work hours as an adaptation mechanism has not been quantified on a global scale. Here we combine heat exposure estimates from reanalysis data¹ with patterns of warming¹⁶ from the latest climate model projections to examine workshift ‘adaptation potential’, or what percent of work time is recoverable if laborers move work hours from the hottest hours

of the day to the cooler morning hours, and how work loss and adaptation potential change as the planet warms.'

To attempt to clarify the potential confusion of 'adaptation' in general with the specific focus of the paper (time shifting as one adaptation mechanism), we have added a sentence to the Introduction (Lines 53-55):

'Although here we define 'adaptation potential' as the ability of workers to shift labor to cooler hours, as mentioned above, adaptation mechanisms are not limited to time shifting.'

We also changed the section heading (Line 138) "Warming impacts on local and global adaptation capacity" to "Warming impacts on local and global potential via time shifting" to remind readers of this point.

Yes, overall adaptive capacity is limited by the limits to the efficacy of this particular option, but it would be good to see some more detailed examination of how that knowledge supports more robust adaptation planning - e.g.

-1: The limits of time/task shifting strategies and what it will cost in terms of lost productivity and GDP, calculated as stand-alone response. This could support arguments for funding alternative adaptation measures (e.g. through Warsaw International Mechanism for Loss and Damage).

- 2: choices between adaptation options, how to assemble a collection of strategies and weigh their utility including over time and under different scenarios (e.g. compensating for lack of value from time shifting by investing in active cooling interventions).

This is a good point. As mentioned above, we added a sentence to the Introduction to emphasize that time shifting is not the only adaptation mechanism. Additionally, in the Discussion we also now mention that future work could consider other adaptation mechanisms, and the other points suggested by the reviewer above (Lines 266-272): 'Additionally, this study has focused on one specific adaptation mechanism, shifting of work to cooler hours. Future work could consider other adaptation strategies, such as task shifting (e.g., movement of labor-intensive tasks to cooler hours), the limits of time and task shifting strategies, what these mechanisms will cost both together and separately in terms of lost productivity and GDP, as well as recommendations for how workers could make choices between adaptation options and weigh their utility under various warming levels and under different scenarios.'

Minor comments -

Line 29 - the grammatical structure of the sentence implies avoiding unsafe working conditions *cause* labour productivity losses directly - whereas the latter usually refers only to a reduction in work rate, which may in fact be a sign

of self-pacing and an appropriate response to heat. Suggest rephrasing (such as " ...unsafe and causes labour productivity losses ..".

We have split this longer sentence into separate sentences to avoid confusion. We now state (Lines 29-31):

'Many low-latitude locations already experience heat exposure that makes physical labor unsafe.^{1,8-10} Labor productivity losses associated with reductions in work rate due to heat exposure can be as high as ...'

Line 42 - Consider costs as well as benefits, to avoid bias to positive outcomes.

We now mention costs as well as benefits in this sentence (Lines 43-45):

'A comprehensive understanding of potential heat exposure and health costs and benefits of shifting work times is needed to weigh trade-offs and to inform decision-making and policies that support adaptation to heat.'

Paragraph from Line 180 - heat strain impairing physical and cognitive function and contributing to higher accident and workplace injury rates is also worth mentioning - e.g. the just published study from Park, Pankratz and Behrer (2021).

We thank the reviewer for pointing out this new work- we now cite it in this sentence (Lines 226-229):

'If laborers are unable to work under safe conditions, they are at higher risk of multiple health impacts, including premature death³²⁻³⁵, workplace injuries³⁶, morbidity from heat-related illness^{37,38}, traumatic injuries^{7,39}, and acute kidney injury³¹.'

269 - A consideration of formal (night) shift work would be valuable to consider in relation to your findings (e.g. swapping the 12 hottest hours for the 12 coolest hours). Large construction sites and resource extraction and processing are examples of heavy labour contexts where night shifts are often used (although usually in the context of 24-hour operations).

As the reviewer has pointed out, moving labor from daylight hours to the night may be possible in some contexts, such as in the construction or resource extraction industries in some locations, where lighting or other logistical issues may present less of an obstacle to adaptation. However, as we note in the Discussion, the ability of workers to move work to the night will be highly context specific, and may be impractical in many contexts given the competing hazards and health risks (e.g., sleep disruption, increased risk of injury, local noise ordinances, lack of lighting, childcare, community impacts, etc.).

Nonetheless, we agree that testing the limit of this adaptation mechanism could be an interesting addition to the paper. Therefore, we have calculated the work losses currently and with 1°C, 2°C, 3°C, and 4°C of additional warming during the 12 hottest hours of the day, as well as during the 12 coolest hours of the day. We find that globally if workers could move labor from the 12 hottest hours of the day to the 12 coolest hours

of the day, they could currently recover ~62% of the lost labor (i.e., 88 billion lost work hours in the coolest 12 hours, and 230 billion lost work hours in the hottest 12 hours); the ability to ‘recover’ lost work time in the 12 coolest hours of the day relative to the 12 hottest hours of the day decreases at a rate of ~3-4%/°C of global warming. For example, under an additional 2°C of warming, only 55% of lost labor productivity can be recovered (251 billion hours lost in the coolest 12 hours, and 554 billion lost hours in the hottest 12 hours)- notably, there is more labor lost in the coolest 12 hours of the day at +2°C global warming than are currently lost in the 12 hottest hours of the day.

We added a brief discussion of these results to the main text, as well as a supplementary figure (Figure S9, also copied below from Lines 176-183):

‘In some industries (e.g., construction, resource extraction) in specific locations, laborers are already able to work at night, so we also assess the ability of workers to recover lost labor by moving work from the 12 hottest hours of the day to the 12 coolest hours of the day. We find that currently, on a global scale, 62% of the labor lost during the 12-hour workday can be recovered by moving the entire shift to the coolest 12 hours of the day. However, global 12-hour work-shift adaptation potential decreases at a rate of about 3-4%/°C of global warming. Notably, under an additional 2°C of future warming, more global labor would be lost in the coolest 12 hours of the day than is currently lost in the hottest 12 hours of the day (Figure S10).’

Figure S10. Global sums of population-weighted heavy labor lost in the hottest 12 hours of the day and in the coolest 12 hours of the day (a) for global-mean temperature changes of 0°C (present day: 2001-2020), +1°C, +2°C, +3°C, and +4°C. Percent of time (b) that can be recovered (or is still lost) by moving labor from the hottest 12 hours of the day to the coolest 12 hours of the day.

From a policy perspective, it is helpful to indicate in the main body/introduction to the paper that analysis of impacts is conservative, given use of ERA5 and basing the analysis on shade conditions underestimates actual WBGT.

We now mention in the Discussion that the use of ERA5 is conservative as it underestimates WBGT, and we point out that assuming workers are in the shade is also conservative (but nonetheless widely used) (Lines 257-259):

'We have also used reanalysis-based hourly estimates of heat exposure, but this method is known to be conservative as it underestimates extremes observed at weather stations¹.'

As well as in the Methods (Lines 305-307):

'It is important to note that WBGT in the sun can be at least 2-3°C higher than shade values⁶⁴, and reanalysis-based estimates of WBGT can underestimate extremes¹, so our estimates of productivity losses from heat exposure may be conservative.'

An indication of future research, and whether your analysis and data are available for use would be helpful.

Upon acceptance of the manuscript, we plan to upload and share netcdf files that include the CMIP6 warming patterns, as well as the estimates of mean heavy labor productivity losses (currently, under 1,2,3,4C warming) on a platform such as Zenodo and have indicated this in the paper.

Similarly, policies that restrict night shift/early morning work such as noise restrictions, industrial zoning etc should not be assumed to be permanent barriers to adaptation, but as triggers for a more extensive investigation of potential adaptation strategies and a reimagining of what a heat-adaptive society might look like.

In the Discussion, we added a sentence that mentions this point (Lines 243-246):

'However, policies that restrict night shift or early morning work such as noise restrictions may not be permanent barriers to adaptation if future investigation of potential adaptation strategies prompt changes in local ordinances to accommodate these strategies.'

The paper already makes a strong and useful argument, and provides valuable findings. With a bit more clarity on the above points it would offer an even more effective launch-pad for further research and more detailed adaptation planning by relevant policy communities.

We again thank the reviewer for their comments- they have been quite helpful. We hope the changes we made addressed their concerns.

Reviewer #3 (Remarks to the Author):

In this manuscript, Parsons et al. develop estimates of global labor productivity loss resulting from humid heat conditions in the present day as well in the future with different amounts of human-induced warming. The authors then quantify the percentage of productivity that could be recovered under several work-shifting scenarios that model changes in when heavy labor is performed within the standard 12-hour workday they modeled.

The questions the authors are asking with this analysis—namely, how will climate change affect the labor productivity of outdoor workers around the world? And how will the capacity to adapt to warming change as that warming grows more severe—are interesting ones. And they are important questions to be considering as nations head into the COP26 international climate negotiations, at which discussions of and commitments to reducing emissions and paying for the costs of climate adaptation take place. Thus, this paper is an important contribution to our collective understanding of the costs of climate change.

Overall, this is a strong piece of research. The manuscript is well written, the methods the authors employed were sound—though there are some comments below that I'd like to see addressed--and their conclusions followed reasonably from their results. Yet neither the methods nor the findings struck me as novel enough to warrant publication in Nature Climate Change given the caliber of the journal. For example, in performing their analysis, the authors essentially employed a previously published methodology for simulating futures with different amounts of warming relative the preindustrial era (i.e., those of Tigchelaar et al. 2020). And the core findings of increasingly severe heat constraints on labor are conceptually similar to the work of Dunne et al. 2013 and Kjellstrom et al. 2018, though it's notable that neither of those previous studies considered potential adaptation measures or adaptation capacity changes as the present manuscript does.

We thank the reviewer for their time and comments.

We believe our analysis represents a new contribution to the literature for several reasons: (1) Although we did follow a similar method to Tigchelaar et al., 2020, our method is distinct in that we use hourly data (Tigchelaar used NARR 3-hourly data), and we account for spatial variations in specific humidity and WBGT in CMIP6 data (Tigchelaar used CMIP5 data, and assumed spatially uniform decreases in relative humidity). (2) As the reviewer points out, although several studies have highlighted labor losses and warming in general, to our knowledge no studies have examined the climatic limits of workshift adaptation and associated changes with warming.

Please see responses and line numbers (line numbers from 'accepted changes' document) below.

Comments:

1. In lines 55-57 and Figure 1, the authors describe locations where humid heat is already at or approaching levels unsafe for continuous heavy labor in the morning and at midday. In looking at the peak WBGT values in Figure 1, however, it's unclear what that unsafe level is and how it relates to the WBGT_{ave} presented in lines 245-246 of the methods section. From the methods section, I assumed that threshold would be 32.47C, but none of the locations in Figure 1 approach that value. I may be misunderstanding, but some clarity around what that threshold WBGT value is and how it relates to the equation in the methods section is needed. I suggest adding a sentence for clarity around lines 55-57.

We thank the reviewer for highlighting this issue if it was unclear- in our initial submission, we did not provide a supplementary figure that showed estimated productivity losses as a function of (s)WBGT- below we copied a new supplementary figure (Figure S1) that shows productivity loss as a function of WBGT. The WBGT_{ave} the reviewer mentions (~32.5°C) is marked with a blue dot in this figure- this dot marks the 50% productivity loss. Very small productivity losses begin at WBGT values ~20°C, with 10% losses at a WBGT value of approximately 27°C.

Figure S1. Fraction of heavy labor productivity lost as a function of heat exposure (Wet Bulb Globe Temperatures, or WBGT) for outdoor workers. This ERF is based on the ERFs used in other recent global labor productivity studies^{6,7}. The red line shows labor productivity loss as a function of heat exposure, the blue dot shows the ‘WBGT average’ value used in the equation shown in the Methods section of the main text, and the horizontal dashed lines denote 10% and 90% productivity losses.

We added the figure above to the Supplement (now Figure S1) and a sentence that highlights key elements of this figure before we begin discussing results in the main text (Lines 66-69):

‘Here we use an exposure response function^{13,25} (ERF) that relates heat exposure to labor productivity losses (Methods). This ERF shows small (<1%) productivity losses at sWBGT of ~20°C, 10% losses at ~27°C, 50% losses at ~32.5°C, and 90% losses at ~38°C (Figure S1).’

2. The authors find that global labor losses due to extreme heat already more than 200 billion hours per year, with greater losses during anomalously or particularly hot years. Given that these findings are based on historical data, the manuscript would be strengthened significantly if they were vetted against an independent source of data. Are there any estimates—global or for any given nation—of actual labor productivity losses due to humid heat over the observational time period? As the authors note in the discussion, outdoor workers often choose to work during the heat despite the health risks because they need the income. And I wonder if the losses calculated here for the historical period reflect what has actually transpired over that time period. Whether vetting the results with an additional data source is possible or not, I'd suggest future exploring in the text how well those historical results are capturing reality.

The reviewer has pointed out an important issue: gridded weather/climate data can provide relatively high-resolution estimates of environmental exposure to heat+humidity for outdoor workers, and exposure response functions (ERFs) can be applied to these estimates of heat exposure, but few 'on the ground' local observations exist of actual work hours lost. In fact, to our knowledge, there are relatively few observation-based, local estimates of the number of annual hours worked (let alone hours lost due to heat exposure) in individual countries: annual data on hours worked are only available for about 40 countries (e.g., OECD estimates: <https://data.oecd.org/emp/hours-worked.htm>). Several isolated studies, such as Masuda et al. (2019), have shown that workers are simply avoiding work during the peak heat of the day, so there is some field-based evidence that work time is already being lost. Takakura et al. (2018, Earth's Future) list several similar field-based studies of outdoor workers in their Table S4, but as Takakura et al. note, there are limited field studies of worker adaptation choices.

Despite the lack of field-based studies confirming the losses found in our study, our method aligns with recent work estimating global-scale labor productivity losses due to heat exposure (e.g., the Kjellstrom et al. ILO report on workers in a warmer world, and the Watts et al. Lancet Countdowns). Watts et al. have estimated recent large-scale labor losses on the order of hundreds of billions of hours/year; their results are similar to the 12-hour workday losses we present in this manuscript (Figure S2).

We agree with the reviewer that more field observations are needed to determine if these global-scale results are accurate. Therefore, we have added a sentence to the Discussion that acknowledges this limitation and suggests this future work (Lines 263-266):

'Although reanalysis-based global estimates of annual heavy labor losses due to heat exposure approach several hundred billion hours per year¹³ (Figure S2), there are relatively few field-based studies that quantify work time lost due to heat exposure^{5,23}; more field observations are needed to verify the results presented here.'

3. Lines 87-88: The authors find a strong relationship between temperature and labor loss, and while the relationship is very clean as presented in Figure 2, it is

largely unsurprising given that the labor loss calculation is directly tied to WBGT as described in the methods. Perhaps some text in this section that more fully describes the importance of this finding would make the findings more compelling.

As the reviewer points out, we describe in the Methods that temperature + humidity are used to calculate WBGT. The new pieces here include: (1) the non-linear increase in labor losses (WBGT increases linearly as the globe warms, but global labor losses increase non-linearly), (2) losses at the coolest hour of the day also increase non-linearly, which has not been examined before, and has important adaptation planning implications. To emphasize these points, we have added a more description to the results (Lines 126-137):

'Air temperature is used in the calculation of sWBGT (Methods), and sWBGT increases relatively linearly as the globe warms^{14,16,17}, but the relationship among global temperatures and labor losses is non-linear. As the globe warms, more land area is exposed to heat exposure that reaches unsafe levels for continuous work (Figure S8), and more hours of the day in warm locations become too hot for continuous labor (Figure 1). Therefore, the relationship among global-mean temperatures and labor grows non-linearly in the coming century. For example, the number of hours lost in the 12-hour workday increases from ~101 billion hours/°C in the last 42 years to 197 billion hours/°C (+/-11 billion hours) with an additional 2°C of global warming (Figure 3). Similarly, productivity losses at the coolest hour of the day also increase non-linearly; losses at the coolest hour of the day increase from ~2.8 billion hours/°C (+/-0.2 billion hours) in the last 42 years to 7.3 billion hours/°C (+/-0.4 billion hours) with 2°C of global warming (Figure 3).'

4. The analyses quantifying labor loss during the hottest and coolest hours of the day seem somewhat arbitrary. Is there evidence to suggest that workers or employers would deliberately shift heavy work in this manner rather than shifting a full workday to cooler hours?

Shifting a full work day to a night shift to avoid heat exposure is an interesting suggestion, but when we searched the literature, we read much more about evidence for shifting a few hours rather than the entire workday. Moving an entire shift of work to the night can prove impractical or pose its own challenges, as we describe in the Discussion (Lines 238-250):

'...moving work to earlier hours may impact sleep duration, which is associated with injury risk^{45,46}. Furthermore, heat exposure can affect sleep⁴⁷, which can affect the risk of injury and heat strain. Approaches to optimize sleep hygiene, and consideration of impacts on circadian rhythms and sleep, should be included in plans to shift work hours. Second, occupations and industries (e.g. construction) in certain settings may be limited in their abilities to shift work hours due to policies such as local noise ordinances⁴⁸. However, policies that restrict night shift or early morning work such as noise restrictions may not be permanent barriers to adaptation if future investigation of potential adaptation strategies prompt changes in local ordinances to accommodate these strategies. Also, changing work hours has the potential to introduce other hazards related to other aspects of ambient conditions such as lighting. These factors should be anticipated and addressed when optimizing work hour timing. Finally, changes in work

schedules need to be coordinated with childcare and other obligations to maintain overall community well-being.'

Therefore, we conclude that shifting all 12 hours of work to the night to avoid heat exposure will be situationally dependent, and could be impractical without large-scale support from employers and communities. Nonetheless, we added an analysis of the 12-hour workshift adaptation potential to the manuscript, (Lines 176-183) and now show these results in a supplemental figure (Figure S10):

'In some industries (e.g., construction, resource extraction) in specific locations, laborers are already able to work at night, so we also assess the ability of workers to recover lost labor by moving work from the 12 hottest hours of the day to the 12 coolest hours of the day. We find that currently, on a global scale, 62% of the labor lost during the 12-hour workday can be recovered by moving the entire shift to the coolest 12 hours of the day. However, global 12-hour work-shift adaptation potential decreases at a rate of about 3-4%/°C of global warming. Notably, under an additional 2°C of future warming, more global labor would be lost in the coolest 12 hours of the day than is currently lost in the hottest 12 hours of the day (Figure S10).'

As it's currently presented, analyzing the potential benefit of shifting one hour of heavy labor to a cooler time of day seems more of a scientific exercise than a practical exploration of how work might actually shift in response to warming temperatures. The shifting of three hours, however, seems a much more likely occurrence. If there's evidence to suggest one-hour shifts in work time are taking place, that would be helpful to include in the section starting on line 166. If not however, I'd recommend trimming the paragraphs on the one-hour shifts and expanding the text on the three-hour shifts in work schedules.

Based on the literature we have encountered, we cannot conclude that shifting 3 hours is more likely than shifting 1-2 hours, and the number of hours a worker can (or is able to) move to earlier hours is situationally dependent. For example, an 'on the ground' study of workers in Indonesia (Masuda et al., 2019) does suggest that during the peak heat of the day, workers are simply choosing to no longer work in open areas/fields. Additionally, Morabito et al. (2020) explicitly tracks the impact of shifting 1h and 2h of work from peak heat to the early morning in Italy and China, suggesting that agricultural workers could regain some lost productivity if they could move these 1-2h of work to the morning: 'The hourly PL in the sun decreased by 2.2% in Guangzhou and 12% in Florence if the workers started working 1 h earlier and even by 9.3% in Guangzhou and 20.2% in Florence if they shifted the working time by 2 h.'

Therefore, our analysis approximately brackets the suggested or observed movement of work times presented in the field-based studies cited above. We now explicitly state the motivation for this analytical choice in the main text before we present results related to movement of work hours or loss of work time at the hottest vs coolest hours of the day (Lines 143-145):

This analytical choice is motivated by a recent study that found Indonesian workers are already avoiding work in the peak heat of the day²³, as well as another study that recommends agricultural workers shift 1-2 hours of work to the early morning to increase productivity²⁴.

5. Lines 257-261 (Methods): The authors assume that the 12-hour workday is evenly split, with four hours at the daily maximum WBGT, four hours at the daily mean WBGT, and four hours at the halfway point between the two. However, the data presented in Figure 1 seem to imply a different distribution of WBGT values over the course of a 12-hour workday. A reference is made to Kjellstrom et al. 2018, but it would be useful to include an explanation of whether or how the 12-hour WBGT data shown in Figure 1 supports this assumption.

To save on computation time and storage space and for better comparison with previously published work, we chose to use the established '4+4+4' method employed in the Lancet and ILO reports, and in the Kjellstrom et al., 2018 paper as the reviewer noted. However, we did compare the work loss results from the 4+4+4 method against losses calculated from the hourly data (12 hour of data, approximately 7AM-7PM) in the locations highlighted in Fig 1 in the main text (Doha, New Delhi, Atlanta). We find that for Atlanta, the difference in work time lost is 1 minute (17 minutes for the hourly data, 16 minutes for the 4+4+4 method), for Doha, the difference is 7 minutes (169 minutes for the hourly data, 176 minutes for the 4+4+4 method), and for New Delhi, the difference is 3 minutes (105 minutes in the hourly data, 108 minutes in the 4+4+4 method). In all cases, the differences are less than ~5% (~5% for Atlanta, ~4% for Doha, and ~3% for New Delhi). Globally, if we use the 4+4+4 method as opposed to the actual hottest 12 hours in the current climate (2001-2020 mean sWBGT), the global sum of heavy labor loss differences are ~1-2% (~228 billion lost hours for the 4+4+4 method, and ~231 billion hours for the hourly data). These differences are much smaller than the interannual variability of hours lost in the 2001-2020 time period (2001-2020 standard deviation of annual sum of global labor lost is ~27 billion hours). Therefore, we conclude that the above methodological differences are relatively minor compared to, for example, uncertainties in the future warming pathway, or year to year variations in climate and heat stress.

We added a sentence to the Methods (Lines 333-336) that directs readers to Text S1: 'Although hourly weather reanalysis data are now available to calculate hourly losses in the 12-hour workday, we have chosen to use the established '4+4+4' method due to computation and data storage constraints and for better comparison with previously published results¹³; further discussion of this method can be found in Text S1.'

And we now describe these differences in supplementary Text S1:

*To estimate labor losses during the 12-hour workday, we calculate the daily mean sWBGT, daily maximum sWBGT, and the halfway point between these two values, and assume 4 hours is spent near each of these values in the 12-hour work day (4*sWBGT max + 4*sWBGT mean + 4*sWBGT half). We compared results from the '4+4+4' approximation against the losses calculated from the hourly sWBGT data (e.g., used the hourly data from the 12-hour workday as opposed to assuming 3 hours is spent at the maximum, minimum, and mean halfway point) for*

the cities shown in Figure 1 and for the global sums of labor lost. We found differences are <5% between the '4+4+4' approximation and the hourly data. For Atlanta the difference in work time lost is 1 minute (17 minutes for the hourly data, 16 minutes for the 4+4+4 method), for Doha the difference is 7 minutes (169 minutes for the hourly data, 176 minutes for the 4+4+4 method), and for New Delhi the difference is 3 minutes (105 minutes for the hourly data, 108 minutes for the 4+4+4 method). Global sums of labor lost similarly show <2% differences between the '4+4+4' method and the hourly data (~228 billion lost hours/year for the '4+4+4' method, and ~231 billion lost hours/year for the hourly data). These global differences among methods are much smaller than the interannual variability of hours lost/year over the 2001-2020 time period (standard deviation: ~27 billion lost hours/year).'

6. Lines 328-331 (Methods): The explanation of why the 1%CO₂ experiment was preferable to the more traditional emissions pathways from CMIP6 was somewhat unclear. The latter is said to have the potential to “create localized differences in the magnitude of warming,” yet the authors then state that what they’re interested in is local temperature changes. If the inclusion of non-CO₂ greenhouse gases causes local temperature changes, would that not be important to include here? I’d like to see a more compelling or more clearly stated justification for using the 1%CO₂ experiment.

We agree that if the 1pctCO₂ vs SSP5-8.5 warming pattern differences were substantial, this should have been highlighted or more thoroughly explained. However, the differences are not robust, nor are they particularly large. Below we have copied Figure S14 for reference. Note that the maximum differences in warming patterns among experiments is <10% (in southern N America, and parts of S and E Asia). Additionally, there is poor model agreement (<75% of CMIP6 models) on the sign of difference in warming patterns among the experiments- the stippling in the difference map shows lack of model agreement over almost all land areas. Therefore, the models don't show good agreement in the sign of difference, and even in locations where they do show agreement, the local differences in the magnitude sWBGT change per degree of global warming are generally quite small relative to the absolute magnitude of change in either experiment. Additionally, as we note in the Methods section, we re-ran our global labor loss calculations using the SSP5-8.5 warming patterns, and the differences are not noticeable.

We have rewritten the relevant paragraph in the Methods (lines 401-406) to clarify the above points:

'We choose the 1%CO₂ experiment because the 21st century Shared Socioeconomic Pathways (SSPs) include highly uncertain, theoretical future transient aerosol, land use, and other forcing changes⁷². To determine if warming patterns among experiments are robust, we compare warming patterns from 1%CO₂ to patterns from the SSP5-8.5⁷⁵ and find that warming patterns are similar (<10% difference in local magnitude), except in isolated locations in the mid-latitude northern hemisphere (see supplementary figures and Text S1).'

We also added an explanation of our reasoning to Text S1 to attempt to shorten the Methods:

‘Warming patterns are generally similar in the idealized 1pctCO2 (Figure 2; Figures S3-S6) and 21st century SSP5-8.5 experiments, with differences in magnitudes of local warming patterns that are <10% (Figure S14). However, in a few isolated northern hemisphere locations, most of the CMIP6 models (>75%) agree that the magnitude of local warming is greater in the SSP5-8.5 experiment than the 1pctCO2 experiment; many of these locations also show large projected decreases in aerosol emissions in the SSP5-8.5 experiment (not shown). Nonetheless, for most land areas there is not good agreement among CMIP6 models in the sign of difference in magnitude of change among these experiments (stippled areas show disagreement in Figure S14), and even in the few locations where the sign is agreed upon, these differences are <10% in the multi-model median. Additionally, a sensitivity test of our future labor loss estimates using warming patterns from the SSP5-8.5 experiment shows no noticeable differences in global labor loss estimates (not shown).’

Figure S14. Warm-season local sWBGT warming patterns per degree of annual-mean, global mean temperature change in 20 1%CO2 and 20 SSP58.5 CMIP6 simulations (yellow-maroon colormap) and warming pattern differences (blue-white-red colormap). On the difference map, red shading indicates higher-magnitude local change in the SSP58.5 experiment, and blue shading indicates higher-magnitude local change in the 1%CO2 experiment. Stippling shows where <75% of CMIP6 models agree on the sign of difference in warming patterns among the experiments. Note that models agree on the sign of change over several mid-latitude northern

hemisphere regions, but these differences are generally <10% relative to the magnitude of change in the 1%CO₂ multi-model median. All necessary variables were not available to calculate sWBGT in the SSP58.5 experiment for the CMIP6 'MIROC6' and 'SAM0-UNICON' models so maps show multi-model medians and differences for 20 of the 22 models shown in Figure S2.

7. Much of the methods section (particularly lines 364-398 or so) describes the temperature and humid heat warming patterns from CMP6 models. The results here are central to the study, as those warming patterns are applied to the reanalysis data in order to simulate future humid heat conditions, and while the choice to unpack them in the methods section is understandable, I'd suggest crafting a paragraph or two describing these findings at a high level and including that text in the main body of the paper.

We appreciate the reviewer's understanding that we wanted to highlight the warming/adaptation piece of the results (which is why we initially put these results in the Methods/Supplement). However, we agree that the warming patterns are central to the adaptation results presented later in the paper. Therefore, we moved the description of the pattern scaling results to the main text, along with what was Figure S2 (the tas, huss, and sWBGT warming patterns). We split Figure S2 into Figure 2 and a supplementary figure (Figure S4) that highlights the tasmax vs tasmin results, and describe these diurnal temperature range results in Text S1 (tasmax and tasmin are not used in the warming patterns, so we felt these results could stay in the Supplement).

The main text now includes a short, new section titled 'Warming patterns in CMIP6 models' (Lines 90-107):

'Coupled Modeling Intercomparison Project, Phase 6 (CMIP6) projections²⁷ show that future sWBGT will increase relatively linearly as the globe warms¹⁷ (Methods). For illustrative purposes, we show the warming patterns for the 75th percentile of daily temperatures, specific humidity, and sWBGT in Figure 2 (warming patterns from individual CMIP6 models shown in Figure S3). Daily mean, maximum, and minimum temperatures over tropical land areas warm at a rate of ~1-1.2°C per degree of global warming (Figure 2a; Figure S4; TextS1). Additionally, CMIP6 models generally agree on the magnitude of specific humidity increases as global temperatures increase, with the fastest increases in humidity across the tropics and the Middle East, South Asia, SE/E Asia, and SE North America (Figure 2b). CMIP6 models also show good agreement in the spatial patterns and magnitudes of local sWBGT changes per degree of global warming (Figure 2c; Figures S5-S6). Specifically, between ~40°N and 40°S, zonal means over land show local changes in sWBGT of ~1-1.2°C per degree of global warming¹⁶, with maxima approaching ~1.5°C per degree of global warming at low latitudes (Figure S6). If we compare the local magnitude of 2-m air temperature and sWBGT warming patterns, we find that air temperature generally rises faster in regions that experience low specific humidity changes per degree of global warming, whereas sWBGT warms at least as quickly as local temperatures in regions with increases in humidity (Figure S7).'

And the Supplement includes (Text S1):

Daily mean, maximum, and minimum temperatures over tropical land areas warm at a rate of ~1-1.2°C per degree of global warming (Figure S4). We are interested in heat exposure at the hottest and coolest hours of the day, so we also examine changes in diurnal temperature range¹ (DTR) per degree of warming by calculating the differences in daily maximum minus minimum warming patterns. Across much of the global land area, there is little agreement among CMIP6 models in the average change in the DTR. However, in parts of the American tropics, southern Europe, and Africa, these models agree on faster warming in daytime maximum relative to minimum temperatures. These models also tend to agree that minimum temperatures will rise more quickly than maximum temperatures in the southern United States and Southwest Asia. Nonetheless, we rely on simulated changes in daily mean temperatures and humidity to calculate future changes in sWBGTT because observed and projected changes in DTR are expected to be small relative to changes in the mean²⁻⁴. We have excluded the CMIP6 NorESM2-LM model from our analysis of tasmax and tasmin in the 1%CO2 experiment because these variables showed unrealistically high and low changes over land areas as compared to the higher-resolution NorESM2-MM and the other CMIP6 models.'

8. Similar to my previous comment, the results described in the methods section seem overly detailed and like they are largely explaining what is presented in the supplementary figures. I'd suggest a) pulling some of this results-focused text from the present methods section into the supplementary information; and b) presenting a few concise metrics that describe these results in the methods section.

We moved several sections of text out of the Methods: (1) we moved the detailed description of the GDP loss calculation methods (Estimating economic impacts of heavy labor productivity losses) to the Supplement, (2) we moved the description of the sensitivity test related to 1pctCO2 vs SSP5-8.5 warming patterns to the Supplement, (3) we moved the pattern scaling descriptions to the main text, (4) we moved the tasmax vs tasmin description to the Supplement, (5) we moved the '4+4+4' vs hourly data sensitivity testing to the Supplement, and (6) we moved the description of the 21st century warming pathways to the Supplement. We hope these edits have shortened the Methods section enough to address the reviewer's concerns.

Reviewer comments, second round review –

Reviewer #1 (Remarks to the Author):

I am happy with the revisions completed, this is a good publication.

Reviewer #2 (Remarks to the Author):

Thank you for the careful and comprehensive consideration of the review comments, in particular the additional analysis of the 12 coolest/hottest hours, which is very interesting. I have no further comments apart from to congratulate the authors on an excellent and very useful paper.

Reviewer #3 (Remarks to the Author):

The authors have done a commendable job revising the manuscript in response to my comments and those of the other reviewers. At this point, I have no further concerns and I look forward to seeing this excellent study published.

REVIEWERS' COMMENTS

No reviewers requested further changes to the manuscript.

Reviewer #1 (Remarks to the Author):

I am happy with the revisions completed, this is a good publication.

We thank the reviewer for their time and supportive comment.

Reviewer #2 (Remarks to the Author):

Thank you for the careful and comprehensive consideration of the review comments, in particular the additional analysis of the 12 coolest/hottest hours, which is very interesting. I have no further comments apart from to congratulate the authors on an excellent and very useful paper.

We thank the reviewer for their time and comments- we agree that the 12 hottest/coolest hours will be of interest to some readers- we are happy to include this in the paper.

Reviewer #3 (Remarks to the Author):

The authors have done a commendable job revising the manuscript in response to my comments and those of the other reviewers. At this point, I have no further concerns and I look forward to seeing this excellent study published.

We thank the reviewer for their time and comments- we are happy to hear that the reviewer is interested in seeing the paper published.